# EHI: End-to-end Learning of Hierarchical Index for Efficient Dense Retrieval

**Ramnath Kumar** *                                        *ramnathk@google.com*
*Google Inc.*

**Anshul Mittal** *†                                       *me@anshulmittal.org*
*Microsoft*
*Google Inc.*

**Nilesh Gupta**                                           *nileshgupta2797@gmail.com*
*UT Austin*
*Google Inc.*

**Aditya Kusupati**                                        *kusupati@google.com*
*Google Inc.*

**Inderjit Dhillon**                                       *isd@google.com*
*Google Inc.*
*UT Austin*

**Prateek Jain**                                           *prajain@google.com*
*Google Inc.*

**Reviewed on OpenReview:** *https://openreview.net/forum?id=GeLLOGsHV9*

## Abstract

Dense embedding-based retrieval is widely used for semantic search and ranking. However, conventional two-stage approaches, involving contrastive embedding learning followed by approximate nearest neighbor search (ANNS), can suffer from misalignment between these stages. This mismatch degrades retrieval performance. We propose End-to-end Hierarchical Indexing (EHI), a novel method that directly addresses this issue by jointly optimizing embedding generation and ANNS structure. EHI leverages a dual encoder for embedding queries and documents while simultaneously learning an inverted file index (IVF)-style tree structure. To facilitate the effective learning of this discrete structure, EHI introduces dense path embeddings that encodes the path traversed by queries and documents within the tree. Extensive evaluations on standard benchmarks, including MS MARCO (Dev set) and TREC DL19, demonstrate EHI's superiority over traditional ANNS index. Under the same computational constraints, EHI outperforms existing state-of-the-art methods by $+\mathbf{1.45\%}$ in MRR@10 on MS MARCO (Dev) and $+\mathbf{8.2\%}$ in nDCG@10 on TREC DL19, highlighting the benefits of our end-to-end approach.

## 1 Introduction

Semantic search (Johnson et al., 2019) aims to retrieve relevant or *semantically similar* documents/items for a given query. In the past few years, semantic search has been applied to numerous real-world applications like web search, product search, and news search (Nayak, 2019; Dahiya et al., 2021). The problem in the

---

*Equal Contribution
†Work done while at Google Inc.

simplest form can be abstracted as: for a given query $q$, retrieve the relevant document(s) $d(q)$ from a static set of documents $\{d_1, d_2, \ldots, d_N\}$ s.t. $d(q) = \arg\max_{1 \leq j \leq N} \text{SIM}(\mathbf{q}, \mathbf{d_j})$. Here $\text{SIM}$ is a similarity function that has high fidelity to the training data $\mathcal{B} = \{(q_i, d_j, y_{ij})\}$. Tuple $(q_i, d_j, y_{ij})$ indicates if document $d_j$ is relevant ($y_{ij} = 1$) or irrelevant ($y_{ij} = -1$) for a given query $q_i \in \mathcal{Q}$.

Dense embedding-based retrieval (Johnson et al., 2019; Jayaram Subramanya et al., 2019; Guo et al., 2020) is the state-of-the-art (SOTA) approach for semantic search and typically follows a two-stage process. In the first stage, it embeds the documents and the query using a deep network like BERT (Devlin et al., 2018) (Additional details about the encoder used in EHI is described in Section 3.2, and the training methodology expanded in Section 3.5). That is, it defines similarity $\text{SIM}(q, d) := \langle \mathcal{E}_\theta(q), \mathcal{E}_\theta(d) \rangle$ as the inner product between embeddings $\mathcal{E}_\theta(q)$ and $\mathcal{E}_\theta(d)$ of the query $q$ and the document $d$, respectively. $\mathcal{E}_\theta(\cdot)$ is a dense embedding function learned using contrastive losses (Ni et al., 2021; Menon et al., 2022).

In the second stage, approximate nearest neighbor search (ANNS) retrieves relevant documents for a given query. That is, all the documents are indexed offline and are then retrieved online for the input query. ANNS in itself has been extensively studied for decades with techniques like ScaNN (Guo et al., 2020), IVF (Sivic & Zisserman, 2003), HNSW (Malkov & Yashunin, 2020), DiskANN (Jayaram Subramanya et al., 2019) and many others being used heavily in practice.

## 1.1 Motivation for EHI

The prevailing two-stage approach to dense retrieval, involving separate training of the embedding encoder and the approximate nearest neighbor search (ANNS) structure, suffers from inherent limitations:

***Embedding-ANNS Misalignment:*** The lack of joint optimization leads to a potential mismatch between the learned embedding space and the requirements of the ANNS structure. This misalignment can cause suboptimal clustering of data points, hindering the ANNS algorithm's ability to efficiently identify semantically similar examples. For example, the documents might be clustered in six clusters to optimize encoder loss. However, due to computational constraints, ANNS might allow only five branches/clusters, thus splitting or merging clusters unnaturally and inaccurately. Figure 1(a) helps illustrate the issue with the disjoint training regime typically used in dense retrieval literature. If the disjoint ANNS tries to cluster these documents into buckets, the resulting model will achieve suboptimal performance since the encoder was never aware of the task of ANNS. Instead, EHI proposes the learning of both the encoder and ANNS structure in a single pipeline. This leads to efficient clustering, as the embeddings being learned already share information about how many clusters the ANNS is hashing.

***Disregard for Query Distribution:*** Conventional ANNS methods focus on generic retrieval efficiency, potentially neglecting the specific characteristics of the query distribution. This can result in an indexing structure that is ill-suited to the patterns present in the actual queries encountered during deployment (Jaiswal et al., 2022). To provide an illustrative instance, let us consider a scenario in which there exists a corpus of documents denoted by $\mathcal{D}$, with two classes of documents categorized as $[D_1, D_2]$, following a certain probability distribution $[d_1, d_2]$. Suppose, $d_1 \gg d_2$, then under the utilization of the unsupervised off-the-shelf ANNS indexer such as ScaNN algorithm, clustering into two partitions, the algorithm would distribute the two document classes in an approximately uniform manner across the resultant clusters. Concurrently, the documents of class $D_1$ would be allocated to both of these clusters, as guided by the partitioning procedure, regardless of the query distribution. However, supposing a scenario in which an overwhelming majority of the queries target $D_2$, the goal emerges to rigorously isolate the clusters associated with $D_1$ and $D_2$ since we would like to improve latency in the downstream evaluation and visit fewer documents for each query. In this context, the EHI emerges as a viable solution as it takes the query distribution into consideration, effectively addressing this concern by facilitating a clear delineation between the clusters corresponding to the distinct domains. Figure 1(b) depicts the query distribution of EHI model trained on the MS MARCO dataset. It is important here to highlight that for tail queries, which are barely searched in practice, EHI is able to grasp the niche concepts and hash as few documents in these buckets as indicated by the very few average number of documents visited. Moving towards more popular queries, or the torso queries, we note a rise in the average number of documents visited as these queries are more broad in general. Moving even further to

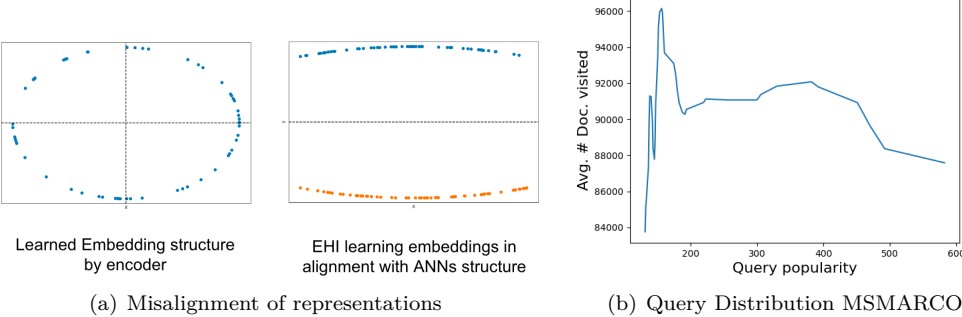

(a) Misalignment of representations       (b) Query Distribution MSMARCO

Figure 1: Motivation of EHI.

the head queries, we again notice a drop in the average number of documents visited, which showcases the efficiency and improved latency of the EHI model.

## 1.2 Overview and Evaluation

Motivated by the aforementioned issues, we propose EHI– **E**nd-to-end learning of **H**ierarchical **I**ndex – that jointly trains both the encoder and the search data structure; see Figure 3. To the best of our knowledge, EHI is the *first* end-to-end learning method for dense retrieval. Similar to two-stage approaches, EHI employs a tree-structured inverted file-like index, but unlike DE+ANNS (the conventional two-stage approach where a standard Dual Encoder (DE) is trained independently, followed by an Approximate Nearest Neighbor Search (ANNS) using the learned embeddings), EHI integrates the training process to simultaneously learn the discrete document assignments within its index and refine the encoder parameters without any warm-starting. We can learn balanced and roughly uniform clusters with just the query, document pairs information which is generally available in the supervised training paradigm of most encoders, and does not require any proxy information through KNN or other clustering algorithms for learning the INF-style tree structure. EHI's core innovation lies in the introduction of dense path embeddings. These embeddings represent the traversal paths of queries and documents within the tree index, enabling the learning of discrete document assignments. By optimizing the index such that semantically similar (query, document) pairs share similar path embeddings, EHI facilitates efficient retrieval by clustering relevant pairs within the same leaf nodes.

We conduct an extensive empirical evaluation of our method against SOTA techniques on standard benchmarks. Our experiments on the popular MS MARCO benchmark (Bajaj et al., 2016) demonstrate that EHI shows improvements of 0.6% in terms of nDCG@10 compared to dense-retrieval with ScaNN baselines when only 10% of documents are searched. We attribute these improved embeddings to the fact that EHI enables an additional signal to the encoder from the indexer as well as the integrated hard negative mining as it can retrieve irrelevant or negative documents from indexed leaf nodes of a query. Here, the indexer parameters are always fresh, unlike techniques akin to ANCE (Xiong et al., 2020). Similarly, EHI provides upto **8.2%** higher nDCG@10 than state-of-the-art (SOTA) baselines on the MS MARCO TREC DL19 (Craswell et al., 2020) benchmarks for the same compute budget. EHI also achieves SOTA exact search performance on both MRR@10 and nDCG@10 metrics with up to **80%** reduction in latency, indicating the effectiveness of the joint learning objective. Similarly, we outperform SOTA architectures such as NCI on NQ320$k$ by **0.6%** and **1.8%** on Recall@10 and Recall@100 metrics. (see Section 4.2).

## 1.3 Contributions

Our work introduces several key advancements to the field of dense retrieval:

- **Novel End-to-End Paradigm.** We propose End-to-end Hierarchical Indexing (EHI), an end-to-end learning framework that jointly optimizes the embedding encoder and the search index structure. EHI addresses the inherent misalignment in conventional two-stage approaches, enabling more efficient and accurate retrieval (Section 3, Figure 2). To the best of our knowledge, we are the first to propose this without any warm starting or soft labels.

- **Rigorous Empirical Validation.** We conduct comprehensive evaluations on the industry-standard MS MARCO benchmark, demonstrating EHI's superiority over state-of-the-art ANNS techniques such as ScaNN, Faiss. Furthermore, we also showcase that EHI is competitive with other SOTA baselines such as ColBERT, SGPT, cpt-text, ANCE, DyNNIBAL. Our experiments underscore EHI's focus on enhancing retrieval accuracy within a fixed computational budget (Section 4.2, Appendix 4.3).
- **Framework Flexibility.** EHI is designed to be agnostic to specific encoder architectures, similarity metrics, and hard negative mining strategies. This flexibility underscores its potential for broad applicability within the dense retrieval domain.

## 2 Related Works

Dense retrieval (Mitra et al., 2018) underlies a myriad of web-scale applications like search (Nayak, 2019), recommendations (Eksombatchai et al., 2018; Jain et al., 2019), and is powered by (a) learned representations (Devlin et al., 2018; Kolesnikov et al., 2020; Radford et al., 2021), (b) ANNS (Johnson et al., 2019; Sivic & Zisserman, 2003; Guo et al., 2020) and (c) LLMs in retrieval (Tay et al., 2022; Wang et al., 2022; Guu et al., 2020). Figure 2 illustrates the difference between dual encoder trained with an ANNS (DE + ANNS), Generative Retrieval approaches (such as DSI, NCI) and EHI.

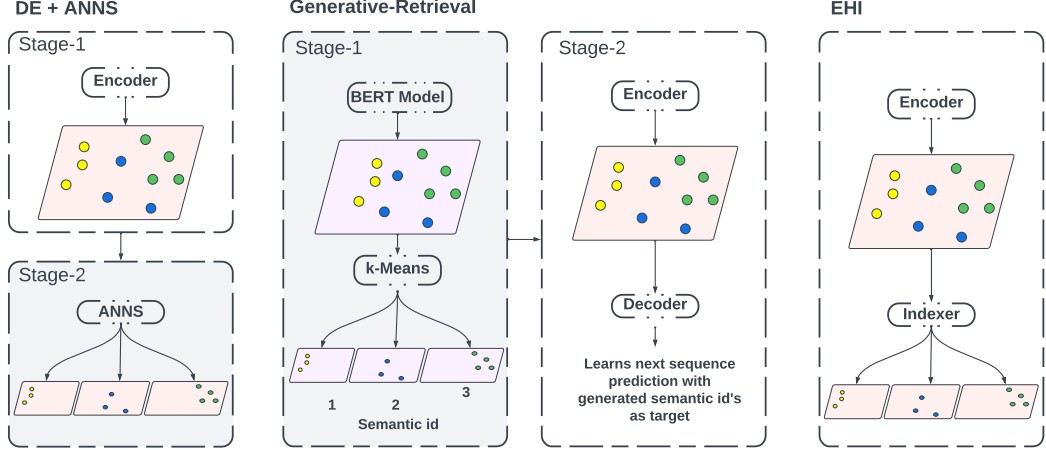

Figure 2: Comparison between EHI's end-to-end training paradigm in comparison to other approaches such as Dual encoder + ANNS and Generative Retrieval approaches which require a multi-stage training.We denote stages which are not optimized for the task objective by a grey shade. Note that unlike DE + ANNS, and Generative Retrieval, EHI optimizes for recall in both encoder and indexer.

**Representation learning.** Powerful representations are typically learned through supervised and un/self-supervised learning paradigms that use proxy tasks like masked language modeling (Devlin et al., 2018) and autoregressive training (Radford et al., 2018). Recent advances in contrastive learning (Gutmann & Hyvärinen, 2010) helped power strong dual encoder-based dense retrievers (Ni et al., 2021; Izacard et al., 2021; Nayak, 2019). They consist of query and document encoders, often shared, which are trained with contrastive learning using limited positively relevant query and document pairs (Menon et al., 2022; Xiong et al., 2020). While most modern-day systems use these learned representations as is for large-scale ANNS, there is no need for them to be aligned with the distance metrics or topology of the data structures. Recent works have tried to address these concerns by warm-starting the learning with a clustering structure (Gupta et al., 2022) but fall short of learning jointly optimized representations alongside the search structure. We explicitly state the challenges of directly adapting methods like ELIAS, originally designed for classification, to the dense retrieval task. However, we include a comparison against ELIAS on a classification benchmark with label features in Appendix C.3. Other works such as RepCONC (Zhan et al., 2022), and SPLADE (Formal et al., 2022) also work on the efficiency aspect of retrieval, where they focus on quantization of the representations using regularizers which explicitly work on reducing FLOPS.

**Approximate nearest neighbor search (ANNS).** The goal of ANNS is to retrieve *almost* nearest neighbors without paying exorbitant costs of retrieving true neighbors (Clarkson, 1994; Indyk & Motwani, 1998; Weber et al., 1998). The "approximate" nature comes from pruning-based search data structures (Sivic & Zisserman, 2003; Malkov & Yashunin, 2020; Beygelzimer et al., 2006) as well as from the quantization based cheaper distance computation (Jegou et al., 2010; Ge et al., 2013). This paper focuses on ANNS data structures and notes that compression is often complementary. Search data structures reduce the number of data points visited during the search. This is often achieved through hashing (Datar et al., 2004; Salakhutdinov & Hinton, 2009; Kusupati et al., 2021), trees (Friedman et al., 1977; Sivic & Zisserman, 2003; Bernhardsson, 2018; Guo et al., 2020) and graphs (Malkov & Yashunin, 2020; Jayaram Subramanya et al., 2019). ANNS data structures also carefully handle the systems considerations involved in a deployment like load-balancing, disk I/O, main memory overhead, etc., and often tree-based data structures tend to prove highly performant owing to their simplicity and flexibility (Guo et al., 2020). For a more comprehensive review of ANNS structures, please refer to Cai (2021); Li et al. (2020); Wang et al. (2021). Works such as CCSA (Lassance et al., 2021) propose alternate ANN structures for efficient retrieval via constrained clustering. Other works such as TDM (Zhu et al., 2018) proposes learning a tree-like ANNS but the ANNS is learnt in a disjoint fashion where k-means algorithm is used for learning the clustering without any supervision. Works such as JTM (Zhu et al., 2019), and Deep Retrieval (Gao et al., 2020b) also attempt to optimize the learn the indexer with a neural network - but they require pre-computation of optimal paths taken by the query and document for learning. Prior works including PQN (Yu et al., 2018) and GPQ (Jang & Cho, 2020) have explored end-to-end ANNS techniques in the visual domain which primarily rely on hashing-based approaches, and not an hierarchical tree-like structure. EHI differentiates itself through its novel tree structure and dense path embeddings, tailored for the unique characteristics of text data and semantic search. Furthermore, EHI handles a significantly larger number of buckets (e.g., 7000) compared to the 48-bit hashing used in approaches like GPQ, demonstrating its suitability for the scale of textual datasets with millions of documents. Similarly, the scale of the experiments are also significantly larger as PQN, GPQ showcase their efficacy on ∼170K images to be indexed (where the overhead of exact search might not be very large), while we experiment on ∼10M documents, where exact search is a significant overhead during inference.

EHI works in an end-to-end fashion without any pre-computation of optimal paths, and is the *first end-to-end hierarchical indexer* framework to the best of our knowledge.

**Generative Retrieval for Semantic Search.** Recently, there have been some efforts towards modeling retrieval as a sequence-to-sequence problem. In particular, Differential Search Index (DSI) (Tay et al., 2022) and more recent Neural Corpus indexer (NCI) (Wang et al., 2022) method proposed encoding the query and then find relevant document by running a learned decoder. However, both these techniques, at their core, use a *separately* computed hierarchical k-means-based clustering of document embeddings for semantically assigning the document-id. That is, they also index the documents using an ad-hoc clustering method which might not be aligned with the end objective of improving retrieval accuracy. In contrast, EHI jointly learns both representation and a k-ary tree-based search data structure end-to-end. This advantage is reflected on MS MARCO dataset EHI is upto 7.12% more accurate (in terms of nDCG@10) compared to DSI. Recently, retrieval has been used to augment LLMs also (Guu et al., 2020; Izacard & Grave, 2020b;a; Izacard et al., 2022). We would like to stress that the goal with LLMs is language modeling while retrieval's goal is precise document retrieval. However, retrieval techniques like EHI can be applied to improve retrieval subcomponents in such LLMs.

## 3 End-to-end Hieararchical Indexing (EHI)

**Problem definition and Notation.** Consider a problem with a corpus of $N$ documents $\mathcal{D} = \{d_1, ..., d_N\}$, a set of $Q$ training queries $\mathcal{Q} = \{q_1, ..., q_Q\}$, and training data $(q_i, d_k, y_{ik})$, where $y_{ik} \in \{-1, 1\}$ is the label for a given training (query, document) tuple and $y_{ik} = 1$ denotes that $d_k$ is relevant to $q_i$. Given these inputs, the goal is to learn a *retriever* that maps a given query to a set of relevant documents while minimizing the computation cost. While wall-clock time is the primary cost metric, comparing different methods against it is challenging due to very different setups (language, architecture, parallelism, etc.). Instead, we rely on recall vs.

% searched curves, widely considered a reasonable proxy for wall-clock time modulo other setup/environment changes (Guo et al., 2020).

## 3.1 Overview of EHI

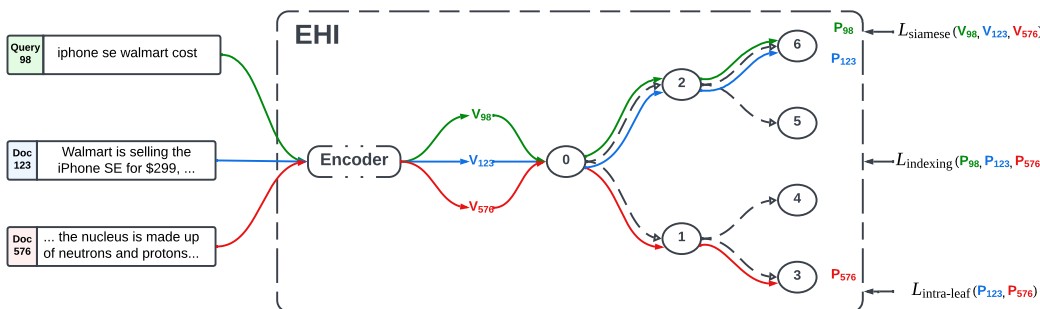

Figure 3: EHI is an end-to-end hierarchical indexer which comprises an encoder and a hierarchical tree as the indexer where the entire pipeline is learnable and differentiable. Here, variables $V_{98}$, $V_{123}$, and $V_{576}$ are dense representations (embeddings) of the text and $P_{98}$, $P_{123}$, and $P_{576}$ are path embeddings of the respective samples. To efficiently train EHI without any warm starting, we use a combination of objectives - $L_{\text{siamese}}$, $L_{\text{indexing}}$, $L_{\text{intra-leaf}}$ (see Section 3 for details).

At a high level, EHI has three key components: Encoder $\mathcal{E}_\theta$, Indexer $\mathcal{I}_\phi$ and Retriever. Parameters $\theta$ of the query/document encoder and $\phi$ of the indexer are the trainable parameters of EHI. Unlike most existing techniques, which train the encoder and indexer in a two-step disjoint process, we train both the encoder and indexer parameters jointly with an appropriate loss function; see Section 3.5. Learning the indexer – generally a discontinuous function – is a combinatorial problem that also requires multiple rounds of indexing the entire corpus. However, by modeling the indexer using a hierarchical tree and its "internal representation" as compressed path embedding, we demonstrate that the training and retrieval with encoder+indexer can be executed efficiently and effectively.

In the following sections, we provide details of the encoder and indexer components. In Section 3.4, we detail how encoder+indexer can be used to retrieve specific documents for a given query, which is used for inference and hard-negative mining (further elaborated in Appendix C.2) during training. Section 3.5 provides an overview of the training procedure. Finally, Section 3.6 summarizes how documents are ranked after retrieval.

## 3.2 Encoder $\mathcal{E}_\theta$: Dense embedding of query/documents

Our method is agnostic to the architecture used for dual encoder. But for simplicity, we use standard dual encoder (Ni et al., 2021) to map input queries and documents to a common vector space. That is, encoder $\mathcal{E}_\theta$ parameterized by $\theta$, maps query ($q \in \mathcal{Q}$) and document ($d \in \mathcal{D}$) to a common vector space: $\mathcal{E}_\theta(q) \in \mathbb{R}^m$, and $\mathcal{E}_\theta(d) \in \mathbb{R}^m$, where $m$ is the embedding size of the model (768 here). While such an encoder can also be multi-modal as well as multi-vector, for simplicity, we mainly focus on standard textual data with single embedding per query/document. We use the standard *BERT architecture* for encoder $\mathcal{E}_\theta$ and initialize parameters $\theta$ using a pre-trained Sentence-BERT distilbert model (Reimers & Gurevych, 2019). Our base model has 6 layers, 768 dimensions, 12 heads with 66 million parameters. We then fine-tune the final layer of the model for the target downstream dataset.

## 3.3 Indexer $\mathcal{I}_\phi$: Indexing of query/document in the hierarchical data structure

EHI's indexer ($\mathcal{I}_\phi$) is a tree with height $H$ and branching factor $B$. Each tree node contains a *classifier* that provides a distribution over its children. So, given a query/document, we can find out the leaf nodes that the query/document indexes into, as well as the *probabilistic* path taken in the tree.

The final leaf nodes reached by the query are essential for retrieval. But, we also propose to use the path taken by a query/document in the tree as an *embedding* of the query/document – which can be used in training through the loss function. However, the path a query/document takes is an object in an exponentially large (in height $H$) vector space, owing to $B^H$ leaf nodes, making it computationally intractable even for a small $H$ and $B$. For instance, with $B = 10$ and $H = 5$, a relatively small tree, there would be 100,000 ($10^5$) possible paths.

Instead, below, we provide a significantly more compressed *path embedding* – denoted by $\mathcal{T}(\cdot; \phi)$ and parameterized by $\phi$ – embeds any given query or document in a relatively low-dimensional $(B \cdot H)$ vector space (which would lead to just 50 in the above example). For simplicity, we denote the query and the document path embedding as $\mathcal{T}_\phi(q) = \mathcal{T}(\mathcal{E}_\theta(q); \phi)$ and $\mathcal{T}_\phi(d) = \mathcal{T}(\mathcal{E}_\theta(d); \phi)$, respectively.

We construct path embedding of a query/document as:

$$\mathcal{T}(\mathcal{E}_\theta(q)) = \mathcal{T}(\mathcal{E}_\theta(q); \phi) = [\mathbf{p}^H; \mathbf{p}^{H-1}; \dots; \mathbf{p}^1],$$

Where $\mathbf{p}^h \in [0, 1]^B$ denotes the probability distribution of children nodes for a parent at height $h$. For a given leaf $l$, say path from root node is defined as $\mathbf{l} = [i_l^1, i_l^2 \dots i_l^H]$ where $i_l^h \in [1 \dots B]$ for $h \in [H]$. The probability at a given height in a path is approximated using a height-specific simple feed-forward neural network parameterized by $\mathbf{W}_{h+1} \in \mathbb{R}^{(B \cdot h + m) \times B}$ and $\mathbf{U}_{h+1} \in \mathbb{R}^{(B \cdot h + m) \times (B \cdot h + m)}$ ($m$ is the embedding size). That is,

$$\mathbf{p}^{h+1} = \texttt{Softmax}(\mathbf{W}_{h+1}^\top \mathcal{K}_h; \mathbf{U}_{h+1})) \cdot \mathbf{p}^h[i_l^h] \tag{1}$$

where $\mathcal{K}_h = \mathcal{F}([\mathbf{o}(i_l^h); \mathbf{o}(i_l^{h-1}); \dots; \mathbf{o}(i_l^1); \mathcal{E}_\theta(q)]$, and one-hot-vector $\mathbf{o}(i)$ is the $i$-th canonical basis vector and $\mathcal{F}$ is a non-linear transformation given by $\mathcal{F}(\mathbf{x}; \mathbf{U}_h) = \mathbf{x} + \texttt{ReLU}(\mathbf{U}_h^\top \mathbf{x})$.

In summary, the path embedding for height 1 represents a probability distribution over the leaves. During training, we compute path embedding for higher heights for only the most probable path, ensuring that the summation of leaf node logits remains a probability distribution. Also, the indexer and path embedding function $\mathcal{T}(\cdot; \phi)$ has the following collection of trainable parameters: $\phi = \{\mathbf{W}_H, \dots, \mathbf{W}_1, \mathbf{U}_H, \dots, \mathbf{U}_1\}$, which we learn by optimizing a loss function based on the path embeddings; see Section 3.5. Please refer to Appendix E for additional details about the indexer in EHI.

### 3.4 Retriever: Indexing items for retrieving

Indexing and retrieval form a backbone for any search structure. EHI efficiently encodes the index path of the query and documents in $(B \cdot H)$-dimensional embedding space. During retrieval for a query q, EHI explores the tree structure to find the "most relevant" leaves and retrieves documents associated with those leaves. For retrieval, it requires encoder and indexer parameters $(\theta, \phi)$ along with Leaf, document hashmap $\mathcal{M}$.

The relevance of a leaf $l$ for a query $q$ is measured by the probability of a query reaching a leaf at height $H$ $(\mathcal{P}(q, l, H))$. Recall from previous section that path to a leaf $l$ is defined as $\mathbf{l} = [i_l^1, i_l^2 \dots i_l^H]$ where $i_l^h \in [1 \dots B]$ for $h \in [H]$. The probability of reaching a leaf $l$ for a given query $q \in \mathcal{Q}$ to an arbitrary leaf $l \in \texttt{Leaves}$ can be computed as $\mathcal{P}(q, l, H) = \mathbf{p}^H[i_l^H]$ using equation 1. But, we only need to compute the most probable leaves for every query during inference, which we obtain by using the standard beam-search procedure summarized below:
1. For all parent node at height $h - 1$, compute probability of reaching their children $\hat{S} = \cup_{c \in \texttt{child}(p)} \mathcal{P}(q, c, h) \; \forall_{p \in P}$
2. Keep top $\beta$ children based on score $\hat{S}$ and designate them as the parents for the next height.
Repeat steps 1 and 2 until the leaf nodes are reached.

Once we select $\beta$ leaves EHI retrieves documents associated with each leaf, which is stored in the hashmap $\mathcal{M}$. To compute this hash map, EHI indexes each document $d \in \mathcal{D}$ (similar to query) with $\beta = 1$. Here, $\beta = 1$ is a design choice considering memory and space requirements and is kept as a tuneable parameter. It stores document-to-leaf associations, allowing us to strategically focus the search on the most relevant subset of documents. The hashmap $\mathcal{M}$ is used to store the information of which documents reach which leaves in our tree. This is essentially the result of the indexing step. Note that to add new ad-hoc documents after

training EHI, one would need to follow the indexing step listed above to store the new documents in the appropriate buckets and update the hashmap $\mathcal{M}$.

During retrieval, we find out which leaves the query goes to, and we only use those documents which have reached similar leaves. This is how we sample a few documents to search over instead of searching over the entire document space. Algorithm 2 in the appendix depicts the approach used by our Indexer for better understanding.

### 3.5  Training EHI

Given the definition of all three EHI components – encoder, indexer, and retriever – we are ready to present the training procedure. As mentioned earlier, the encoder and the indexer parameters $(\theta; \phi)$ are optimized simultaneously with our proposed loss function, which is designed to have the following properties: a) Relevant documents and queries should be semantically similar, b) documents and queries should be indexed together iff they are relevant, and c) documents should be indexed together iff they are similar.

Given the encoder and the indexer, we design one loss term for each of the properties mentioned above and combine them to get the final loss function. To this end, we first define the triplet loss as:

$$L(a, b, c) = [ac - ab + \gamma]_+ \qquad (2)$$

where we penalize if similarity between query $q$ and an *irrelevant* document $d_-$ ($y(q, d_-) \neq 1$) is within $\gamma$ margin of the corresponding similarity between $q$ and a relevant document $d_+$ ($y(q, d_+) = 1$).We now define the following three loss terms:

1. **Semantic Similarity**: the first term is a standard dual-encoder contrastive loss between a relevant document $d_+$ – i.e., $y(q, d_+) = +1$ – and an *irrelevant* document with $y(q, d_-) \neq 1$.

$$L_{\text{siamese}} = L(\mathcal{E}_\theta(q), \mathcal{E}_\theta(d_+), \mathcal{E}_\theta(d_-); \theta) \qquad (3)$$

2. **Indexing Similarity**: the second term is essentially a similar contrastive loss over the query, relevant-doc, irrelevant-doc triplet, but where the query and documents are represented using the path-embedding $\mathcal{T}_\phi(\cdot)$ given by the indexer $\mathcal{I}_\phi$.

$$L_{\text{indexing}} = L(\mathcal{T}_\phi(q), \mathcal{T}_\phi(d_+), \mathcal{T}_\phi(d_-); \theta, \phi) \qquad (4)$$

3. **Intra-leaf Similarity**: to spread out irrelevant docs, third loss applies triplet loss over the sampled relevant and irrelevant documents for a query $q$. Note that we apply the loss only if the two docs are semantically dissimilar according to the latest encoder, i.e., $\texttt{SIM}(\mathbf{a}, \mathbf{b}, \tau) = \frac{\mathcal{E}_\theta(a)^T \mathcal{E}_\theta(b)}{|\mathcal{E}_\theta(a)||\mathcal{E}_\theta(b)|} < \tau$ for a pre-specified threshold $\tau = 0.9$.

$$L_{\text{intra-leaf}} = \texttt{SIM}(d_+, d_-, \tau) \mathcal{T}_\phi(d_+)^\top \mathcal{T}_\phi(d_-) \qquad (5)$$

The final loss function $\mathcal{L}$ is given as the weighted sum of the above three losses:

$$\mathcal{L} = \lambda_1 L_{\text{siamese}} + \lambda_2 L_{\text{indexing}} + \lambda_3 L_{\text{intra-leaf}} \qquad (6)$$

Here $\gamma$ is set to 0.3 for all loss components, and $\lambda_1, \lambda_2, \lambda_3$ are tuneable hyper-parameters. Our trainer (see Algorithm 1) learns $\theta$ and $\phi$ by optimizing $\mathcal{L}$ using standard techniques; for our implementation we used AdamW (Loshchilov & Hutter, 2017). Note that the loss function only uses in-batch documents' encoder embeddings and path embeddings, i.e., we are not even required to index all the documents in the tree structure, thus allowing efficient joint training of both encoder and indexer. To ensure fast convergence, we use hard negatives mined from the indexed leaves of a given query $q$ for which we require documents to be indexed in the tree. But, this procedure can be done once in every $r$ step where $r$ is a hyper-parameter set to 5 by default across our experiments. We will like to stress that existing methods like DSI, NCI, or ANCE not only have to use stale indexing of documents, but they also use stale or even fixed indexers – like DSI, NCI learns a fixed semantic structure over docids using one-time hierarchical clustering. In contrast, EHI jointly updates the indexer and the encoder in each iteration, thus can better align the embeddings with the tree/indexer.

### 3.6 Re-ranking and Evaluation

This section describes the re-ranking step and the test-time evaluation process after retrieval. In Section 3.4, we discussed how each document is indexed, and we now have a learned mapping of $d \times l$, where $d$ is the corpus size, and $l$ is the number of leaves. Given a query at test time, we perform a forward pass similar to the indexing pipeline presented in Section 3.4 and find the top-$b$ leaves ($b$ here is the beam size) the given query reaches. We collate all the documents that reached these $b$ leaves (set operation to avoid any repetition of the same documents across multiple leaves) and rank them based on an appropriate similarity metric such as cosine similarity, dot product, manhattan distance, etc. We use the cosine similarity metric for ranking throughout our experiments (see Section 4.2).

## 4 Experiments

We conduct a comprehensive empirical evaluation of EHI on established dense retrieval benchmarks. Our experiments have two primary objectives:

(a) **Validate the End-to-end Paradigm:** We seek to demonstrate the superiority of EHI's joint optimization of the embedding encoder and Indexer compared to the conventional disjoint training approach that utilizes off-the-shelf ANNS methods (e.g., ScaNN, Faiss-IVF) or alternative generative retrieval or sparse retrieval approaches. This comparison provides strong evidence for the benefits of our proposed paradigm shift.

(b) **Justify Design Decisions:** Through systematic experimentation, we aim to elucidate the rationale behind the design choices within EHI. These insights will solidify the theoretical foundations of our method and guide future research in this domain.

Detailed training hyperparameters for EHI are provided in Appendix B.

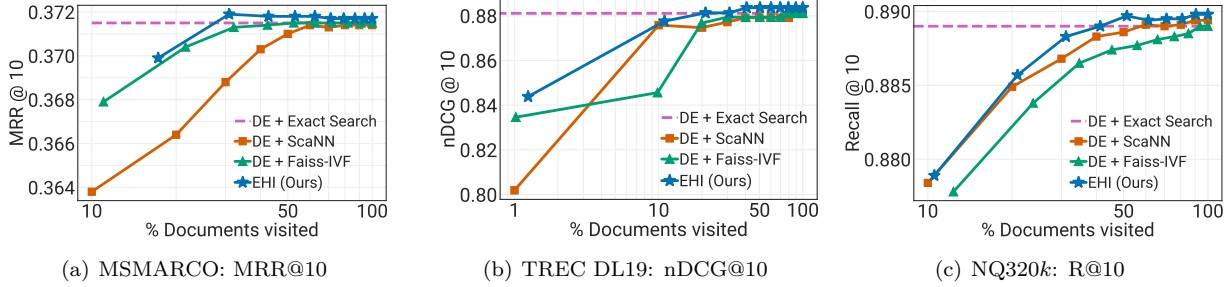

(a) MSMARCO: MRR@10      (b) TREC DL19: nDCG@10      (c) NQ320$k$: R@10

Figure 4: EHI is significantly more accurate than DE + ScaNN or Faiss-IVF, especially when restricted to visit a small fraction of documents. See Figure 10 in Appendix for results on Scifact, Fiqa.

### 4.1 Experimental Setup

**Datasets.** We evaluate EHI on four standard but diverse retrieval datasets of increasing size: SciFact (Wadden et al., 2020), FIQA (Maia et al., 2018), NQ320$k$ (Kwiatkowski et al., 2019), and MS MARCO (Bajaj et al., 2016). Appendix A provides additional details about these datasets.

We establish a robust set of baselines to comprehensively evaluate EHI's performance and isolate the impact of our core contributions:

**State-of-the-Art (SOTA) Comparisons.** We evaluate EHI against leading dense retrieval methods such as ColBERT-v2 (Santhanam et al., 2021), SPLADE (Formal et al., 2022), and many other baselines on the MS MARCO benchmark (see Table 1, Section 4.3). These comparisons, while acknowledging architectural differences, demonstrate EHI's strong performance and its potential within the broader research landscape. Importantly, EHI achieves this competitiveness using a significantly small encoder (66M parameters), highlighting the effectiveness of our end-to-end index learning approach. We note that re-ranking and other late interaction methods employed in ColBERT-v2 and SPLADE currently outperform EHI in exact search

scenarios on the MS MARCO benchmark. This presents a promising future research direction to explore how re-ranking losses and other techniques could be integrated into EHI. Note that methods like ColBERT-v2, due to their late-interaction similarity computation, are less readily adaptable to ANNS-based retrieval.

**Controlled Ablation Studies.** To directly assess the value of EHI's end-to-end indexer, we construct baselines using the same DistilBERT encoder paired with: (a) Dual Encoder (DE with ANCE hard-negatives; Menon et al. (2022)) + Exact Search (for theoretical upper bound), (b) DE + ScaNN (representative of popular ANNS libraries; Guo et al. (2020)), and (c) DE + Faiss-IVF (for comparison with another IVF-style index; Johnson et al. (2019)).

**Generative Retrieval.** We include baselines like DSI (Tay et al., 2022) and NCI (Wang et al., 2022) to provide context, acknowledging their potential limitations in this setting. We report DSI numbers on MS MARCO using an implementation validated by the authors. However, we note that NCI fails to scale to large datasets like MS MARCO.

**Sparse Neural Retrieval.** We include sparse neural IR baselines like CCSA (Lassance et al., 2021) and SPLADE (Formal et al., 2022) for a comprehensive comparison in Table 1. However, it's important to note their potential differences in our context. Methods like CCSA, with their two-step disjoint training, face inherent challenges similar to those discussed earlier. While SPLADE offers notable improvements by replacing ANNS with sparse representations and using a FLOP-minimizing regularization term, this approach necessitates training multiple models for different FLOP targets of the practitioner. In contrast, EHI achieves this flexibility and elasticity within a single model. Furthermore, as Lassance et al. (2023) observes, SPLADE models can exhibit slower inference speeds compared to dense IR models which could be alleviated with static pruning, but the earlier limitation still holds water. However, the distillation-based improvements achieved by SPLADE are promising, and exploring the integration of distillation techniques with EHI represents an exciting direction for future research.

## 4.2 Results

Evaluations and findings on Scifact and Fiqa datasets is presented in Appendix C. We evaluate EHI on the MS MARCO passage retrieval task, a *widely recognized benchmark* for semantic search. Results are presented for both the standard dev set and the TREC DL-19 set (Craswell et al., 2020). We also report our findings on other datasets such as NQ320*k* (Kwiatkowski et al., 2019). We compare against the standard Sentence-BERT model (Huggingface, 2019), trained on the MS MARCO dataset, which is further fine-tuned to the various datasets compared in this paper.

**Efficiency-Accuracy Trade-off:** EHI demonstrates a compelling advantage in the efficiency-accuracy trade-off. For instance, on the standard MS MARCO dev set, EHI matches or exceeds the accuracy of Exact Search while visiting **80%** fewer documents (see Figure 4(a)). This significantly outperforms DE+ScaNN and DE+Faiss-IVF, which require visiting nearly double the documents for comparable accuracy. Furthermore, on the TREC-DL 19 benchmark, EHI is able to match or surpass the nDCG@10 of baseline Exact Search with an **78%** reduction in latency (see Figure 4(b)). On the NQ320*k* dataset, EHI matches or surpasses the accuracy of baseline Exact Search with a **60%** reduction in latency (see Figure 4(c)).

**Gains in compute bottlenecks:** We emphasize the significance of EHI's **3.6%** nDCG@10 improvement (with only 1% documents visited) on the well-optimized MS MARCO benchmark. Such gains underscore the practical value of our approach. Similarly, when restricted to visiting 1% of the documents, EHI achieves **8.2%** higher nDCG@10 than DE+ScaNN and DE+Faiss-IVF on the TREC-DL benchmark.

**Comparison with Generative Retrieval Methods:** We note that the DSI base model with 250M parameters is almost *four times* the size of the current EHI model. After multiple weeks of DSI training with doc2query + atomic id + base model, DSI achieves a MRR@10 metric of 26% on the MS MARCO dev set, which is **8.58%** lower than EHI with just **1**% visited documents (see Table 1). Note that despite significant efforts, we could not scale NCI code (Wang et al., 2022) on MS MARCO due to the dataset size; NCI paper does not provide metrics on MS MARCO dataset. To compare against NCI, we report our findings on the NQ320*k* dataset. Note that EHI is able to significantly outperform DSI and NCI (without query generation)

Table 1: Performance metrics (%) evaluated on the MS MARCO dev dataset. The best value for each metric is indicated in **bold** font.

| Method | MRR@10 (Dev) | nDCG@10 (Dev) | #. parameters |
|---|---|---|---|
| DPR Thakur et al. (2021) | - | 17.7 | 110M (BERT-base) |
| BM25 (official) Khattab & Zaharia (2020) | 16.7 | 22.8 | - |
| BM25 (Anserini) Khattab & Zaharia (2020) | 18.7 | - | - |
| cpt-text S Neelakantan et al. (2022) | 19.9 | - | 300M (cpt-text S) |
| cpt-text M Neelakantan et al. (2022) | 20.6 | - | 1.2B (cpt-text M) |
| cpt-text L Neelakantan et al. (2022) | 21.5 | - | 6B (cpt-text L) |
| cpt-text XL Neelakantan et al. (2022) | 22.7 | - | 175B (cpt-text XL) |
| DSI (Atomic Docid + Doc2Query + Base Model) Tay et al. (2022) | 26.0 | 32.28 | 250M (T5-base) |
| DSI (Naive String Docid + Doc2Query + XL Model) Tay et al. (2022) | 21.0 | - | 3B (T5-XL) |
| DSI (Naive String Docid + Doc2Query + XXL Model) Tay et al. (2022) | 16.5 | - | 11B (T5-XXL) |
| DSI (Semantic String Docid + Doc2Query + XL Model) Tay et al. (2022) | 20.3 | 27.86 | 3B (T5-XL) |
| CCSA Lassance et al. (2021) | 28.9 | - | 110M (BERT-base) |
| HNSW Malkov & Yashunin (2018) | 28.9 | - | - |
| RoBERTa-base + In-batch Negatives Monath et al. (2023) | 24.2 | - | 123M (RoBERTa-base) |
| RoBERTa-base + Uniform Negatives Monath et al. (2023) | 30.5 | - | 123M (RoBERTa-base) |
| RoBERTa-base + DyNNIBAL Monath et al. (2023) | 33.4 | - | 123M (RoBERTa-base) |
| RoBERTa-base + Stochastic Negatives Monath et al. (2023) | 33.1 | - | 123M (RoBERTa-base) |
| RoBERTa-base +Negative Cache Monath et al. (2023) | 33.1 | - | 123M (RoBERTa-base) |
| RoBERTa-base + Exhaustive Negatives Monath et al. (2023) | 34.5 | - | 123M (RoBERTa-base) |
| SGPT-CE-2.7B Muennighoff (2022) | - | 27.8 | 2.7B (GPT-Neo) |
| SGPT-CE-6.1B Muennighoff (2022) | - | 29.0 | 6.1B (GPT-J-6B) |
| SGPT-BE-5.8B Muennighoff (2022) | - | 39.9 | 5.8B (GPT-J) |
| doc2query Khattab & Zaharia (2020) | 21.5 | - | - |
| DeepCT Thakur et al. (2021) | 24.3 | 29.6 | 110M (BERT-base) |
| docTTTTquery Khattab & Zaharia (2020) | 27.7 | - | - |
| SPARTA Thakur et al. (2021) | - | 35.1 | 110M (BERT-base) |
| docT5query Thakur et al. (2021) | - | 33.8 | - |
| ANCE | 33.0 | 38.8 | - |
| EHI (distilbert-cos; **Ours**) | **33.8** | **39.4** | **66M** (distilBERT) |
| RepCONC Zhan et al. (2022) | 34.0 | - | 123M (RoBERTa-base) |
| TAS-B Thakur et al. (2021) | - | 40.8 | 110M (BERT-base) |
| GenQ Thakur et al. (2021) | - | 40.8 | - |
| ColBERT (re-rank) Khattab & Zaharia (2020) | 34.8 | - | 110M(BERT-base) |
| ColBERT (end-to-end) Khattab & Zaharia (2020) | 36.0 | 40.1 | 110M (BERT-base) |
| BM25 + CE Thakur et al. (2021) | - | 41.3 | - |
| SPLADE (simple training) Formal et al. (2022) | 34.2 | - | 110M (BERT-base) |
| SPLADE + DistilMSE Formal et al. (2022) | 35.8 | - | 66M (distilBERT) |
| SPLADE + SelfDistil Formal et al. (2022) | 36.8 | - | 66M (distilBERT) |
| SPLADE + EnsembleDistil Formal et al. (2022) | 36.9 | - | 66M (distilBERT) |
| EHI (distilbert-dot; **Ours**) | 37.2 | **43.3** | **66M** (distilBERT) |
| SPLADE + CoCondenser-SelfDistil Formal et al. (2022) | 37.5 | - | 66M (distilBERT) |
| SPLADE + CoCondenser-EnsembleDistil Formal et al. (2022) | 38.0 | - | 66M (distilBERT) |
| ColBERT-v2 Santhanam et al. (2021) | **39.7** | - | 110M (BERT-base) |

despite NCI utilizing a $10\times$ larger encoder! Furthermore, even with query generation, NCI is 0.6% and 1.8% less accurate than EHI on Recall@10 and Recall@100 metrics, respectively. (see Table 3)

Our observations about EHI are statistically significant as evidenced by p-value tests Section 4.5. Additional experiments such as the results on other benchmarks such as scifact, fiqa, negative mining, comparison against Elias (Gupta et al., 2022), and qualitative analysis on the leaves of the indexer learned by the EHI model is depicted in Appendix C.

## 4.3 Comparisons to SOTA

In this section, we compare EHI with SOTA approaches such as ColBERT, SGPT, cpt-text, ANCE, DyN-NIBAL, which use different dot-product metrics or hard-negative mining mechanisms, etc. Note that EHI proposed in this paper is a paradigm shift of training and can be clubbed with any of the above architectures. So the main comparison is not with SOTA architectures, but it is against the existing paradigm of two-stage dense retrieval as highlighted in Section 4.2. Nevertheless, we showcase that EHI outperforms SOTA approaches on both MS MARCO dev set as well as MS MARCO TREC DL-19 setting as highlighted in Table 1 and Table 2 respectively. Table 3 depicts the performance of EHI in comparison to other SOTA approaches.

Table 2: Performance metrics (%) evaluated on the MS MARCO TREC-DL 2019 Craswell et al. (2020) dataset. The best value for each metric is indicated in **bold** font.

| Method | MRR@10 | nDCG@10 | #. parameters |
|---|---|---|---|
| BM25 Robertson et al. (2009) | 68.9 | 50.1 | - |
| CCSA Lassance et al. (2021) | - | 58.3 | 110M (BERT-base) |
| RepCONC Zhan et al. (2022) | 66.8 | - | 123M (RoBERTa-base) |
| Cross-attention BERT (12-layer) Menon et al. (2022) | 82.9 | 74.9 | 110M (BERT-base) |
| Dual Encoder BERT (6-layer) Menon et al. (2022) | 83.4 | 67.7 | 66M (distilBERT) |
| DistilBERT + MSE Menon et al. (2022) | 78.1 | 69.3 | 66M (distilBERT) |
| SPLADE (simple training) Formal et al. (2022) | - | 69.9 | 110M (BERT-base) |
| DistilBERT + Margin MSE Menon et al. (2022) | 86.7 | 71.8 | 66M (distilBERT) |
| DistilBERT + RankDistil-B Menon et al. (2022) | 85.2 | 70.8 | 66M (distilBERT) |
| DistilBERT + Softmax CE Menon et al. (2022) | 84.6 | 72.6 | 66M (distilBERT) |
| DistilBERT + $M^3$SE Menon et al. (2022) | 85.2 | 71.4 | 66M (distilBERT) |
| SPLADE + DistilMSE Formal et al. (2022) | - | 72.9 | 66M (distilBERT) |
| SPLADE + SelfDistil Formal et al. (2022) | - | 72.3 | 66M (distilBERT) |
| SPLADE + EnsembleDistil Formal et al. (2022) | - | 72.1 | 66M (distilBERT) |
| SPLADE + CoCondenser-SelfDistil Formal et al. (2022) | - | 73.0 | 66M (distilBERT) |
| SPLADE + CoCondenser-EnsembleDistil Formal et al. (2022) | - | 73.2 | 66M (distilBERT) |
| EHI (distilbert-cos; **Ours**) | **97.7** | **88.4** | **66M** (distilBERT) |

Table 3: Performance metrics (%) evaluated on the NQ320*k* dataset (Kwiatkowski et al., 2019). The best value for each metric is indicated in **bold** font.

| Method | R@10 | R@100 | #. parameters |
|---|---|---|---|
| BM25 Robertson et al. (2009) | 32.48 | 50.54 | - |
| BERT + ANN (Faiss) Johnson et al. (2019) | 53.63 | 73.01 | 110M (BERT-base) |
| BERT + BruteForce Wang et al. (2022) | 53.42 | 73.16 | 110M (BERT-base) |
| BM25 + DocT5Query Nogueira et al. (2019) | 61.83 | 76.92 | - |
| ANCE (FirstP) Xiong et al. (2020) | 80.33 | 91.78 | - |
| ANCE (MaxP) Xiong et al. (2020) | 80.38 | 91.31 | - |
| SEAL (Base) Bevilacqua et al. (2022) | 79.97 | 91.39 | 139M (BART-base) |
| SEAL (Large) Bevilacqua et al. (2022) | 81.24 | 91.93 | 406M (BART-large) |
| RoBERTa-base + In-batch Negatives Monath et al. (2023) | 69.5 | 84.3 | 123M (RoBERTa-base) |
| RoBERTa-base + Uniform Negatives Monath et al. (2023) | 72.3 | 84.8 | 123M (RoBERTa-base) |
| RoBERTa-base + DyNNIBAL Monath et al. (2023) | 75.4 | 86.2 | 123M (RoBERTa-base) |
| RoBERTa-base + Stochastic Negatives Monath et al. (2023) | 75.8 | 86.2 | 123M (RoBERTa-base) |
| RoBERTa-base +Negative Cache Monath et al. (2023) | - | 85.6 | 123M (RoBERTa-base) |
| RoBERTa-base + Exhaustive Negatives Monath et al. (2023) | 80.4 | 86.6 | 123M (RoBERTa-base) |
| DSI (Base) Tay et al. (2022) | 56.6 | - | 250M (T5-base) |
| DSI (Large) Tay et al. (2022) | 62.6 | - | 3B (T5-XL) |
| DSI (XXL) Tay et al. (2022) | 70.3 | - | 11B (T5-XXL) |
| NCI (Base) Wang et al. (2022) | 85.2 | 92.42 | 220M (T5-base) |
| NCI (Large) Wang et al. (2022) | 85.27 | 92.49 | 770M (T5-L) |
| NCI *w/ qg-ft* (Base) Wang et al. (2022) | 88.48 | 94.48 | 220M (T5-base) |
| NCI *w/ qg-ft* (Large) Wang et al. (2022) | 88.45 | 94.53 | 770M (T5-L) |
| EHI (distilbert-cos; **Ours**) | **88.98** | **96.3** | **66M** (distilBERT) |

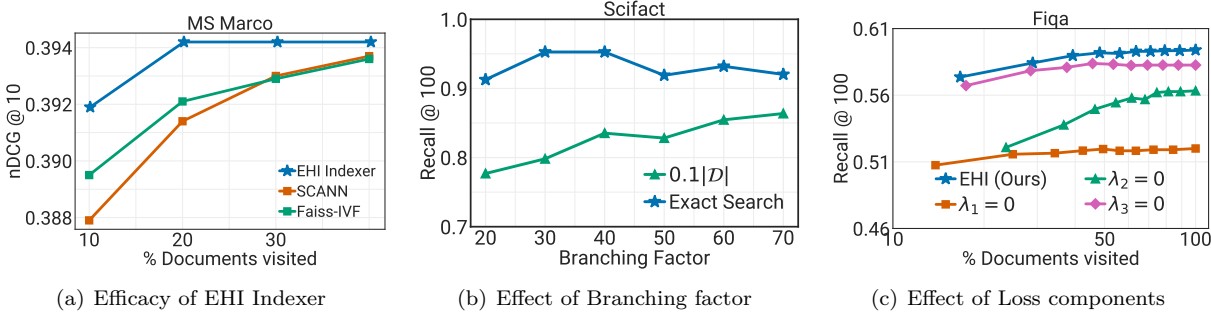

Figure 5: Ablations studies depicting the various properties of EHI's training paradigm.

## 4.4 Robustness to initializations

In this section, we showcase the robustness of our proposed approach - EHI to various initializations of the model. Note that analysis in this manner is uncommon in efficient retrieval works, as showcased in ScaNN, DSI, and NCI. However, such approaches worked with two disjoint processes for learning embeddings, followed by the ANNS. Since we attempt to solve the problem in an end-to-end framework in a differentiable fashion through neural networks, it is worth understanding the effect of different initializations or seeds on our learning. Overall, we run the same model for the best hyperparameters over five different seeds and report the mean and standard deviation numbers below in Table 4.

Table 4: Performance metrics (%) evaluated on the various datasets across various ranges of visited documents. The best value for each metric, corresponding to a specific number of visited documents, is indicated in **bold** font. Here, $n$ denotes the number of seeds run for the given model. We report the mean and standard deviation of the metrics for EHI and showcase a robustness to initialization.

| Metric | Method | 10% | 20% | 30% | 40% | 50% | 60% | 70% | 80% | 90% | 100% |
|---|---|---|---|---|---|---|---|---|---|---|---|
| Scifact - R@100 | Exact Search | | | | | 91.77 | | | | | |
| | ScaNN | 67.89 | 80.76 | 83.76 | 87.49 | 89.82 | 90.6 | 91.6 | 91.27 | 91.27 | 91.77 |
| | Faiss-IVF | 71.81 | 79.42 | 83.93 | 86.93 | 89.6 | 91.1 | 91.77 | 91.77 | 91.77 | 91.77 |
| | EHI ($n=5$) | **80.68** $_{\pm 1.51}$ | **87.05** $_{\pm 1.92}$ | **90.06** $_{\pm 0.97}$ | **91.75** $_{\pm 0.89}$ | **93.03** $_{\pm 0.77}$ | **93.26** $_{\pm 0.72}$ | **93.34** $_{\pm 0.73}$ | **93.44** $_{\pm 0.62}$ | **93.51** $_{\pm 0.62}$ | **93.51** $_{\pm 0.62}$ |
| Fiqa - R@100 | Exact Search | | | | | 53.78 | | | | | |
| | ScaNN | 51.0 | 51.9 | 52.33 | 52.69 | 52.63 | 53.39 | 52.49 | 52.97 | 53.64 | 53.78 |
| | Faiss-IVF | 52.75 | 53.27 | 53.65 | 53.65 | 53.65 | 53.63 | 53.78 | 53.78 | 53.78 | 53.78 |
| | EHI ($n=5$) | **57.23** $_{\pm 0.42}$ | **57.74** $_{\pm 0.27}$ | **57.91** $_{\pm 0.27}$ | **57.98** $_{\pm 0.26}$ | **58.07** $_{\pm 0.25}$ | **58.08** $_{\pm 0.24}$ | **58.11** $_{\pm 0.21}$ | **58.11** $_{\pm 0.21}$ | **58.1** $_{\pm 0.22}$ | **58.1** $_{\pm 0.22}$ |
| MS Marco - nDCG@10 | Exact Search | | | | | 39.39 | | | | | |
| | ScaNN | 38.61 | 39.1 | 39.25 | 39.3 | 39.37 | 39.36 | 39.38 | 39.38 | 39.39 | 39.39 |
| | Faiss-IVF | 38.86 | 39.09 | 39.23 | 39.31 | 39.35 | 39.36 | 39.36 | 39.38 | 39.38 | 39.39 |
| | EHI ($n=5$) | **39.17** $_{\pm 0.04}$ | **39.34** $_{\pm 0.03}$ | **39.39** $_{\pm 0.02}$ | **39.41** $_{\pm 0.01}$ | **39.41** $_{\pm 0.01}$ | **39.42** $_{\pm 0.01}$ | **39.42** $_{\pm 0.01}$ | **39.42** $_{\pm 0.01}$ | **39.42** $_{\pm 0.01}$ | **39.42** $_{\pm 0.01}$ |
| MSMarco - MRR@10 | Exact Search | | | | | 33.77 | | | | | |
| | ScaNN | 33.16 | 33.58 | 33.67 | 33.7 | 33.76 | 33.75 | 33.77 | 33.77 | 33.77 | 33.77 |
| | Faiss-IVF | 33.34 | 33.54 | 33.66 | 33.72 | 33.75 | 33.75 | 33.74 | 33.76 | 33.76 | 33.77 |
| | EHI ($n=5$) | **33.62** $_{\pm 0.03}$ | **33.74** $_{\pm 0.03}$ | **33.77** $_{\pm 0.02}$ | **33.78** $_{\pm 0.01}$ | **33.78** $_{\pm 0.01}$ | **33.79** $_{\pm 0.01}$ | **33.79** $_{\pm 0.01}$ | **33.79** $_{\pm 0.01}$ | **33.79** $_{\pm 0.01}$ | **33.79** $_{\pm 0.01}$ |

## 4.5 Statistical Significance Tests

In this section, we aim to show that the improvements achieved by EHI over other baselines as depicted in Section 4.2 are statistically significant. We confirm with hypothesis tests that all the results on datasets considered in this work are statistically significant with a p-value less than 0.05.

Table 5: Statistical Significance p-test.

| Dataset | Scifact (R@100) | Fiqa (R@100) | MSMARCO (nDCG@10) | MSMARCO (MRR@10) |
|---|---|---|---|---|
| p-value | 0.0016 | 7.97E-07 | 0.0036 | 0.0018 |

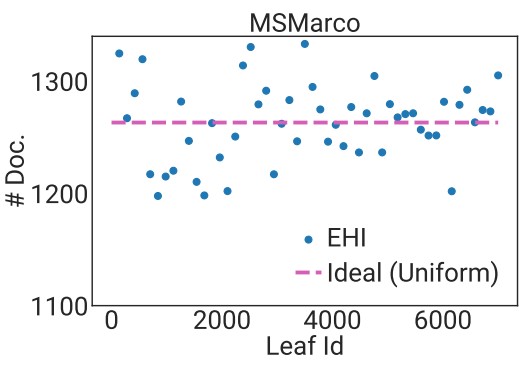

(a) MS MARCO: Document load balancing

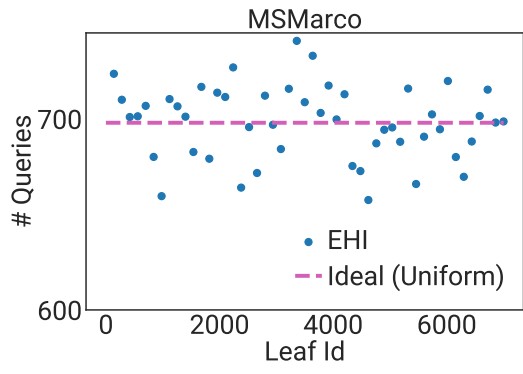

(b) MS MARCO: Query load balancing

Figure 6: Leaf distribution of documents and queries of MS Marco in our trained EHI model. Our model learns splits closer to ideal split (uniform), this suggests that all the leaves are efficiently used, and our model does not learn very lob sided clusters. This roughly uniform spread of both queries (Figure 6(b)), as well as documents (Figure 6(a)) provides a more fine-grained clustering structure which not only takes advantage of all the leaves but also shows state of the art performance compared to baselines as discussed extensively in Section 4.2.

## 4.6 Load balancing

Load balancing is a crucial objective that we aim to accomplish in this work. By achieving nearly uniform clusters, we can create clusters with finer granularity, ultimately reducing latency in our model. In Figure 6(a), we can observe the distribution of documents across different leaf nodes. It is noteworthy that the expected number of documents per leaf, calculated as $\sum_{i=1}^{l=\#\text{Leaf}} p_i c_i$, where $p_i$ represents the probability of a document in bucket index $i$ and $c_i$ denotes the number of documents in bucket $i$, yields an optimal value of approximately 1263.12. Currently, our approach attains an expected number of documents per leaf of 1404.65. Despite a slight skewness in the distribution, we view this as advantageous since we can allocate some leaves to store infrequently accessed tail documents from our training dataset. Additionally, our analysis demonstrates that queries are roughly evenly divided, highlighting the successful load balancing achieved, as illustrated in Figure 6.

## 4.7 Learning to balance load over epochs

In the previous section, we showed that our learned clusters are almost uniformly distributed, and help improve latency as shown in Section 4.2. In this section, we study the progression of expected documents per leaf as defined above over training progression on the Scifact dataset. In Figure 7 demonstrates the decline of the expected documents per leaf over the training regime.

## 4.8 Discussions

In the previous section, we demonstrated the effectiveness of EHI against multiple baselines on diverse benchmarks. In this section, we report results from multiple ablation studies to better understand the behavior of EHI.

**Fraction of visited documents as a proxy for latency.** Ideally, we would want to compare query throughput against recall/MRR, but obtaining head-to-head latency numbers is challenging as different systems are implemented using different environments and optimizations (for instance, off-the-shelf indexers such as SCANN, FAISS work better with CPU optimization while EHI takes advantage of GPUs). Thus, following standard practice in the ANNS community, we use a fraction of documents visited/searched as a proxy for latency (Jayaram Subramanya et al., 2019; Guo et al., 2020).

**Stand-alone Indexer comparisons.** To showcase the alignment between the embeddings and the downstream clusters learned, we conduct an experiment where we initialize the encoders with the encoder trained

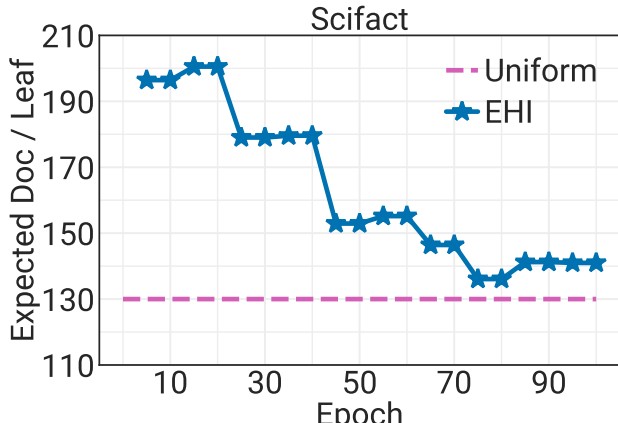

Figure 7: Load balancing of documents over the training regime of EHI. EHI learning objective encourages well balances leaf nodes as well as state of the art accuracy.

via EHI. We find the indexer trained by EHI is better aligned with the representations and leads to better performance when visiting fewer fraction of documents (see Figure 8). This underscores the crucial alignment and efficiency gains achieved through EHI's joint optimization.

To isolate the EHI indexer's impact, we fix encoder embeddings (pre-trained EHI model) and compare EHI indexing against ScaNN, and Faiss-IVF. Figure 5(a) illustrates that EHI outperforms other baselines by up to **4.6%** when visiting a limited number of documents, highlighting the EHI indexer's intrinsic efficiency.

**Effect of branching factor.** Figure 5(b) shows recall@100 of EHI on SciFact with varying branching factors. We consider two versions of EHI, one with exact-search, and another where where we restrict EHI to visit about 10% visited document. Interestingly, for EHI + Exact Search, the accuracy decreases with a higher branching factor, while it increases for the smaller beam-size of 0.1. We attribute this to documents in a leaf node being very similar to each other for high branching factors (fewer points per leaf). We hypothesize that EHI is sampling highly relevant documents for hard negatives leading to a lower exact search accuracy.

**Ablation w.r.t loss components.** Next, on FIQA dataset, we study performance of EHI when one of the loss components equation 6 is turned off; see Figure 5(c). First, we observe that EHI outperforms the other three vanilla variants, implying that each loss term contributes non-trivially to the performance of EHI. Next, we observe that removing the document-similarity-based loss term ($\lambda_3$), Eq. equation 5, has the least effect on the performance of EHI, as the other two-loss terms already capture some of its desired consequences. However, turning off the contrastive loss on either encoder embedding ($\lambda_1$), Eq. equation 3, or path embedding ($\lambda_2$), Eq. equation 4, loss leads to a significant loss in accuracy. This also indicates the importance of jointly and accurately learning encoder and indexer parameters.

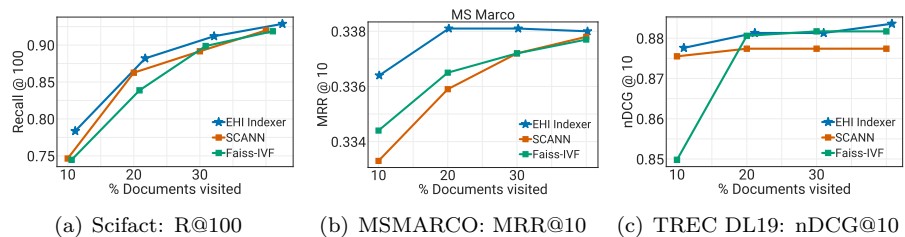

(a) Scifact: R@100    (b) MSMARCO: MRR@10    (c) TREC DL19: nDCG@10

Figure 8: EHI indexer is significantly more accurate than EHI encoder + off-the-shelf indexers such as ScaNN, Faiss-IVF especially when restricted to visit a small fraction of documents. Note that all the algorithms have been initialized with the same EHI encoder for computing embeddings.

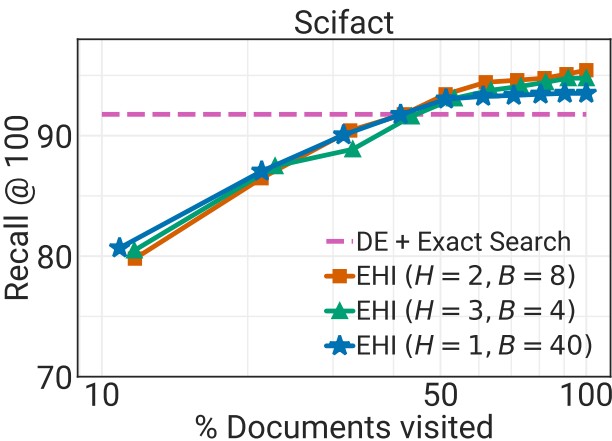

Figure 9: EHI upon extending to deeper trees.

## 4.9 Scaling to deeper trees

In this section, we further motivate the need to extend to deep trees and showcase how this extension is critical to be deployed in production. We show that the hierarchical part in EHI can be tailored for large-scale tasks (>1B documents or otherwise) and is absolutely necessary for two reasons:

**Sub-optimal convergence and blow-up in path embedding:** When working with hundreds of billions of scale datasets as ones generally used in industry, a single height indexer would not work. Having more than a billion logits returned by a single classification layer would be suboptimal and lead to poor results. This is a common trend observed in other domains, such as Extreme Classification, and thus warrants hierarchy in our model design. Furthermore, the path embedding would blow up if we use a single height indexer, and optimization and gradient alignments would become the bottleneck in this scenario.

**Sub-linear scaling and improved latency:** Furthermore, increasing the height of EHI can be shown to scale to very large leaves and documents. Consider a case where our encoder has $P$ parameters, and the indexer is a single height tree that tries to index each document of $D$ dimension into $L$ leaves. The complexity of forward pass in this setting would be $\mathcal{O}(P + DL)$. However, for very large values of $L$, this computation could blow up and lead to suboptimal latency and blow up in memory. Note, however, that by extending the same into multiple leaves ($H$), one can reduce the complexity at each height to a sublinear scale. The complexity of forward pass the hierarchical k-ary tree would be $\mathcal{O}(P + HDL^{1/H})$. Note that since $L \gg H$, the hierarchical k-ary tree can be shown to have a better complexity and latency, which helps in scaling to documents of the scale of billions as observed in general applications in real-world scenarios. For instance, EHI trained on Scifact with equal number of leaves, we notice a significant speedup with increasing height; for example at ($B = 64, H = 1$), ($B = 8, H = 2$), and ($B = 4, H = 3$), we notice a per-query latency of $2.48ms$, $2.40ms$, and $1.99ms$ respectively at the same computation budget.

We study the accuracy of EHI when extending to multiple heights of the tree structure to extend its effectiveness in accurately indexing extensive web-scale document collections. Traditional indexing methods that rely on a single-height approach can be computationally impractical and sub-optimal when dealing with billions or more documents. To address this challenge, EHI treats height as a hyperparameter and learns the entire tree structure end-to-end. This extension to hierarchical k-ary tree is absolutely necessary for scalability.

In the above results presented in Figure 9, we showcase that one could increase the depth of the tree to allow multiple linear layers of lower logits - say 1000 clusters at each height instead of doing so at a single level. This is precisely where the hierarchical part of the method is **absolutely necessary** for scalability. Furthermore, we also extend EHI model on other datasets such as Scifact to showcase this phenomenon as depicted in Table 6. We do confirm with a p-test with a p-value of 0.05 and confirm that the results across different

permutations are nearly identical, and the improvements on R@50 and R@100 over other permutations are not statistically significant (improvements $< 0.5\%$). Furthermore, improvements of other permutations of (B,H) on the R@10 and R@20 metrics over the H=1 baseline are also statistically insignificant. Our current experiments on Scifact/MSMARCO not only portray that our method does indeed scale to deeper trees but also showcase that EHI can extend to deeper trees with little to no loss in performance and improved latency. We also maintain a roughly similar performance and uniform splitting of documents even when extended to deeper trees as depicted in Figure 9.

Table 6: Performance metrics (%) evaluated on the Scifact dataset across various permutations of braching factor (B) and height (H). The best value for each metric, corresponding to a specific number of visited documents, is indicated in **bold** font.

| Scifact | R@10 | R@20 | R@50 | R@100 |
|---|---|---|---|---|
| Two-tower model (with Exact Search) | 75.82 | 82.93 | 87.9 | 91.77 |
| EHI (B=40, H=1) | 84.73 | 87.57 | 93.87 | 95.27 |
| EHI (B=4, H=2) | 86.23 | 88.7 | 93.43 | 94.77 |
| EHI (B=6, H=2) | 83.8 | 88.63 | 92.77 | 95.27 |
| EHI (B=8, H=2) | 84.46 | 88.47 | 93.27 | 95.27 |
| EHI (B=4, H=3) | 84.96 | 88.7 | 93.6 | 94.77 |
| EHI (B=6, H=3) | 85.36 | 89.3 | 93.6 | 94.77 |
| EHI (B=8, H=3) | 83.86 | 87.47 | 92.27 | 95.27 |

## 5 Conclusions, Limitations, and Future Work

We presented EHI, a novel framework that fundamentally shifts the paradigm of dense retrieval by jointly optimizing the query/document encoder and the search indexer. EHI's core components – the encoder, indexer, and retriever – work in tandem to generate compact path embeddings. These path embeddings, representing the path taken by queries and documents within the index tree, are crucial for EHI's effective joint training. Extensive evaluations on standard benchmarks demonstrate EHI's superior efficiency and accuracy compared to traditional approaches. While the success of path embeddings highlights their potential, a deeper theoretical understanding and formal guarantees would be valuable. Additionally, exploring the integration of EHI with hierarchical representations like Matryoshka Embeddings (Kusupati et al., 2022) or techniques like RGD (Kumar et al., 2023), and extending to billion scale (see Appendix D) could further enhance performance, particularly for complex and tail queries, and be of interest to the broader research community. We believe EHI lays a strong foundation for future innovations in dense retrieval, promising more efficient and accurate semantic search systems.

## Broader Impact Statement

Our proposed approach helps improve large scale retrieval and has potential to be applied to multiple industry grade retrieval solutions. Improving scale and recall quality of retrieval itself might have multiple societal conquences, but noe of which are specific to our method.

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

# A   Datasets

In this section, we briefly discuss the open-source datasets used in this work adapted from the Beir benchmark Thakur et al. (2021).

**Scifact:**   Scifact (Wadden et al., 2020) is a fact-checking benchmark that verifies scientific claims using evidence from research literature containing scientific paper abstracts. The dataset has $\sim 5000$ documents and has a standard train-test split. We use the original publicly available dev split from the task having 300 test queries and include all documents from the original dataset as our corpus. The scale of this dataset is smaller and included to demonstrate even splits and improvement over baselines even when the number of documents is in the order of 5000.

**Fiqa:**   Fiqa (Maia et al., 2018) is an open-domain question-answering task in the domain of financial data by crawling StackExchange posts under the Investment topic from 2009-2017 as our corpus. It consists of $57,368$ documents and publicly available test split from (Thakur et al., 2021) As the test set, we use the random sample of 500 queries provided by Thakur et al. (2021). The scale of this dataset is 10x higher than Scifact, with almost 57,638 documents in our corpus.

**MS Marco:**   The MSMarco benchmark Bajaj et al. (2016) has been included since it is widely recognized as the gold standard for evaluating and benchmarking large-scale information retrieval systems (Thakur et al., 2021; Ni et al., 2021). It is a collection of real-world search queries and corresponding documents carefully curated from the Microsoft Bing search engine. What sets MSMarco apart from other datasets is its scale and diversity, consisting of approximately 9 million documents in its corpus and 532,761 query-passage pairs for fine-tuning the majority of the retrievers. Due to the increased complexity in scale and missing labels, the benchmark is widely known to be challenging. The dataset has been extensively explored and used for fine-tuning dense retrievers in recent works (Thakur et al., 2021; Nogueira & Cho, 2019; Gao et al., 2020a; Qu et al., 2020). MSMarco has gained significant attention and popularity in the research community due to its realistic and challenging nature. Its large-scale and diverse dataset reflects the complexities and nuances of real-world search scenarios, making it an excellent testbed for evaluating the performance of information retrieval algorithms.

**NQ320$k$:**   The NQ320$k$ benchmark (Kwiatkowski et al., 2019) has become a standard information retrieval benchmark used to showcase the efficacy of various SOTA approaches such as DSI (Tay et al., 2022) and NCI (Wang et al., 2022). In this work, we use the same NQ320$k$ preprocessing steps as NCI. The queries are natural language questions, and the corpus is Wikipedia articles in HTML format. During preprocessing, we filter out tags and other special characters and extract the title, abstract, and content strings. Please refer to Wang et al. (2022) for additional details about this dataset.

Table 7 presents more information about our datasets.

Table 7: Dataset details.

| Dataset | # Train Pairs | # Test Queries | # Test Corpus | Avg. Test Doc per Query |
|---------|---------------|----------------|---------------|-------------------------|
| Scifact | 920 | 300 | 5183 | 1.1 |
| Fiqa-2018 | 14,166 | 648 | 57,638 | 2.6 |
| MS Marco | 532,761 | 6,980 | 8,841,823 | 1.1 |

# B   Model and Hyperparameter details

We followed a standard approach called grid search to determine the optimal hyperparameters for training the EHI model. This involved selecting a subset of the training dataset and systematically exploring different combinations of hyperparameter values that perform best on this held-out set. The key hyperparameters we

focused on were the Encoder ($\mathcal{E}_\theta$ ) and Indexer ($\mathcal{I}_\phi$) learning rates. Other hyperparameters to tune involved the weights assigned to each component of the loss function, denoted as $\lambda_1, \lambda_2, \lambda_3$ in Equation 6.

The specific hyperparameter values used in our experiments are detailed in Table 8. This table provides a comprehensive overview of the exact settings for each hyperparameter. The transformer block used in this work is the sentence transformers distilBERT model, which has been trained on the MSMarco dataset ([https://huggingface.co/sentence-transformers/msmarco-distilbert-cos-v5](https://huggingface.co/sentence-transformers/msmarco-distilbert-cos-v5)). We fine-tune the encoder for our task and report our performance metrics.

Table 8: Hyperparameters used for training EHI on various datasets. Note that number of epoch and refresh rate ($r$) was set to 100 and 5, respectively. EHI initializes with distilBERT with encoder embedding representation as 768

| Description | Scifact | Fiqa | MSMarco | NQ320k | argument_name |
|---|---|---|---|---|---|
| *General Settings* | | | | | |
| Batch size | 64 | 64 | 4096 | 1024 | batch_size |
| Number of leaves | 40 | 5000 | 7000 | 1000 | leaves |
| *Optimizer Settings* | | | | | |
| Encoder Learning rate | $4e^{-4}$ | $2e^{-4}$ | $2.5e^{-8}$ | $1e^{-4}$ | enc_lr |
| Classifier Learning rate | 0.016 | $9e^{-4}$ | $8e^{-3}$ | $4e^{-4}$ | cl_lr |
| Loss factor 1 | 0.2 | 0.03 | 0.11 | 0.2 | $\lambda_1$ |
| Loss factor 2 | 0.8 | 0.46 | 0.84 | 0.4 | $\lambda_2$ |
| Loss factor 3 | 0.2 | 0.54 | 0.60 | 0.11 | $\lambda_3$ |

# C   Additional Experiments

This section presents additional experiments conducted on our proposed model, referred to as EHI. The precise performance metrics of our model, as discussed in Section 4.2, are provided in Appendix C.1. This analysis demonstrates that our approach achieves state-of-the-art performance even with different initializations, highlighting its robustness (Appendix 4.4).

Furthermore, we delve into the document-to-leaves ratio concept, showcasing its adaptability to degrees greater than one and the advantages of doing so, provided that our computational requirements are met (Appendix C.4). This flexibility allows for the exploration of more nuanced clustering possibilities. We also examine load balancing in EHI and emphasize the utilization of all leaves in a well-distributed manner. This indicates that our model learns efficient representations and enables the formation of finer-grained clusters (Appendix 4.6).

To shed light on the learning process, we present the expected documents per leaf metric over the training regime in Appendix 4.7. This analysis demonstrates how EHI learns to create more evenly distributed clusters as training progresses, further supporting its effectiveness.

Finally, we provide additional insights into the semantic analysis of the Indexer in Appendix C.4, highlighting the comprehensive examination performed to understand the inner workings of our model better.

Through these additional experiments and analyses, we reinforce the efficacy, robustness, and interpretability of our proposed EHI model, demonstrating its potential to advance the field of information retrieval.

**SciFact.** We first start with the small-scale SciFact dataset. Figure 10(a) and Table 9 compares EHI to three DE baselines. Clearly, EHI's recall-compute curve dominates that of DE+ScaNN and DE+Faiss-IVF. For example, when allowed to visit/search about 10% of documents, EHI obtains up to **+15.64%** higher Recall@100. Furthermore, EHI can outperform DE+Exact Search with a **60%** reduction in latency. Finally, representations from EHI's encoder with exact search can be as much as **4%** more accurate (in terms of Recall@100) than baseline dual-encoder+Exact Search, indicating effectiveness of EHI's integrated hard negative mining.

Table 9: Performance metrics (%) evaluated on the Scifact dataset across various ranges of visited documents. The best value for each metric, corresponding to a specific number of visited documents, is indicated in **bold** font.

| Metric | Method | 10% | 20% | 30% | 40% | 50% | 60% | 70% | 80% | 90% | 100% |
|---|---|---|---|---|---|---|---|---|---|---|---|
| R@10 | Exact Search | | | | | 82.73 | | | | | |
| | ScaNN | 66.2 | 74.11 | 78.28 | 79.94 | 81.28 | 81.61 | 81.94 | 82.28 | 82.28 | 82.28 |
| | Faiss-IVF | 70.74 | 78.27 | 80.07 | 82.23 | 82.9 | 82.73 | 82.73 | 82.73 | 82.73 | 82.73 |
| | EHI | **76.97** | **81.93** | **83.33** | **83.67** | **84.73** | **85.07** | **84.73** | **84.73** | **84.73** | **84.73** |
| R@20 | Exact Search | | | | | 86.9 | | | | | |
| | ScaNN | 68.32 | 77.97 | 0.8247 | 84.63 | 85.97 | 86.3 | 86.63 | 86.97 | 86.97 | 86.97 |
| | Faiss-IVF | 73.26 | 81.43 | 84.07 | 86.73 | 87.4 | 87.23 | 86.9 | 86.9 | 86.9 | 86.9 |
| | EHI | **81.03** | **85.27** | **86.5** | **86.17** | **87.23** | **87.57** | **87.57** | **87.57** | **87.57** | **87.57** |
| R@50 | Exact Search | | | | | 92.53 | | | | | |
| | ScaNN | 72.39 | 82.7 | 87.2 | 90.2 | 91.53 | 91.87 | 92.2 | 92.53 | 92.53 | 92.53 |
| | Faiss-IVF | 76.66 | 85.4 | 87.37 | 90.03 | 91.03 | 90.87 | 90.87 | 90.87 | 90.87 | 90.87 |
| | EHI | **82.87** | **88.67** | **91.73** | **92.07** | **93.2** | **93.53** | **93.53** | **93.53** | **93.87** | **93.87** |
| R@100 | Exact Search | | | | | 94.1 | | | | | |
| | ScaNN | 72.62 | 83.03 | 88.53 | 91.53 | 92.87 | 93.2 | 93.53 | 93.87 | 93.87 | 93.87 |
| | Faiss-IVF | 77.16 | 87.3 | 89.93 | 92.6 | 93.27 | 93.43 | 93.43 | 93.77 | 94.1 | 94.1 |
| | EHI | **83.53** | **89.33** | **92.9** | **93.23** | **94.43** | **94.77** | **94.77** | **94.93** | **95.27** | **95.27** |

**FIQA.** Here also we observe a similar trend as SciFact; see Figure 10(b) and Table 10. That is, when restricted to visit only 15% documents (on an average), EHI outperforms ScaNN and Faiss-IVF in Recall@100 metric by **5.46%** and **4.36%** respectively. Furthermore, EHI outperforms the exact search in FIQA with a **84%** reduction in latency or documents visited. Finally, when allowed to visit about 50% of the documents, EHI is about **5%** more accurate than Exact Search, which visits *all* the documents. Thus indicating better quality of learned embeddings.

## C.1 Results on various benchmarks

In this section, we depict the precise numbers we achieve with various baselines as well as EHI as shown in Table 9, Table 10, and Table 11 for Scifact, Fiqa, and MS Marco respectively.

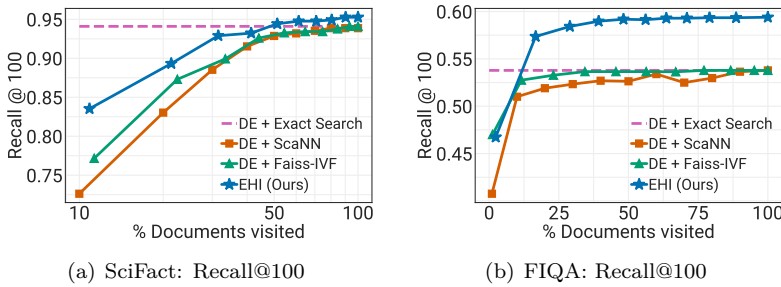

(a) SciFact: Recall@100        (b) FIQA: Recall@100

Figure 10: (a), (b): Recall@100 of EHI and baselines on Scifact and Fiqa dataset. EHI is significantly more accurate than dual encoder + ScaNN or Faiss-IVF baselines, especially when computationally restricted to visit a small fraction of documents. For example, on Scifact dataset, at 10% document visit rate, EHI has 11.7% higher Recall@100 than baselines.

To understand how the performance of EHI model is on a smaller number of $k$ in Recall@$k$, we report Recall@10, Recall@25, and Recall@50 metrics on the Scifact dataset. Figure 11 depicts the performance of the EHI model in this setting. On the Recall@10, Recall@20 and Recall@50 metric, EHI outperforms exact search metric by 8.91%, 4.64% and 5.97% respectively.

Table 10: Performance metrics (%) evaluated on the Fiqa dataset across various ranges of visited documents. The best value for each metric, corresponding to a specific number of visited documents, is indicated in **bold** font.

| Metric | Method | 10% | 20% | 30% | 40% | 50% | 60% | 70% | 80% | 90% | 100% |
|---|---|---|---|---|---|---|---|---|---|---|---|
| **R@10** | Exact Search | | | | | 30.13 | | | | | |
| | ScaNN | 23.95 | 29.08 | 29.69 | 29.98 | 29.83 | 30.13 | 30.08 | 30.08 | 30.13 | 30.13 |
| | Faiss-IVF | 26.8 | 29.67 | 29.9 | 29.93 | 29.93 | 29.93 | 29.98 | 30.13 | 30.13 | 30.13 |
| | EHI | **27.43** | **32.02** | **32.12** | **32.19** | **32.42** | **32.42** | **32.42** | **32.42** | **32.47** | **32.47** |
| **R@20** | Exact Search | | | | | 36.0 | | | | | |
| | ScaNN | 28.54 | 34.45 | 35.34 | 35.67 | 35.7 | 36.0 | 35.88 | 35.95 | 36.0 | 36.0 |
| | Faiss-IVF | 32.17 | 35.51 | 35.77 | 35.8 | 35.8 | 35.8 | 35.85 | 36.0 | 36.0 | 36.0 |
| | EHI | **33.47** | **39.01** | **39.36** | **39.45** | **39.68** | **39.68** | **39.68** | **39.68** | **39.73** | **39.73** |
| **R@50** | Exact Search | | | | | 45.4 | | | | | |
| | ScaNN | 35.93 | 43.38 | 44.51 | 45.11 | 45.02 | 45.12 | 45.07 | 45.48 | 45.5 | 45.4 |
| | Faiss-IVF | 39.78 | 44.45 | 44.86 | 45.19 | 45.19 | 45.19 | 45.24 | 45.4 | 45.4 | 45.4 |
| | EHI | **41.03** | **49.77** | **50.35** | **50.62** | **50.75** | **50.83** | **50.99** | **50.99** | **51.04** | **51.04** |
| **R@100** | Exact Search | | | | | 53.78 | | | | | |
| | ScaNN | 51.0 | 51.9 | 52.33 | 52.69 | 52.63 | 53.39 | 52.49 | 52.97 | 53.64 | 53.78 |
| | Faiss-IVF | 52.75 | 53.27 | 53.65 | 53.65 | 53.65 | 53.63 | 53.78 | 53.78 | 53.78 | 53.78 |
| | EHI | **57.36** | **58.42** | **58.97** | **59.18** | **59.13** | **59.29** | **59.29** | **59.34** | **59.34** | **59.39** |

Table 11: Performance metrics (%) evaluated on the MS Marco dataset across various ranges of visited documents. The best value for each metric, corresponding to a specific number of visited documents, is indicated in **bold** font.

| Metric | Method | 10% | 20% | 30% | 40% | 50% | 60% | 70% | 80% | 90% | 100% |
|---|---|---|---|---|---|---|---|---|---|---|---|
| **nDCG@1** | Exact Search | | | | | 22.41 | | | | | |
| | ScaNN | 22.06 | 22.38 | 22.36 | 22.36 | 22.41 | 22.39 | 22.41 | 22.41 | 22.41 | 22.41 |
| | Faiss-IVF | 22.16 | 22.32 | 22.38 | 22.41 | 22.41 | 22.41 | 22.39 | 22.39 | 22.39 | 22.41 |
| | EHI | **22.39** | **22.48** | **22.46** | **22.46** | **22.46** | **22.46** | **22.46** | **22.46** | **22.46** | **22.46** |
| **nDCG@3** | Exact Search | | | | | 32.5 | | | | | |
| | ScaNN | 31.96 | 32.36 | 32.42 | 32.43 | 32.49 | 32.48 | 32.5 | 32.5 | 32.5 | 32.5 |
| | Faiss-IVF | 32.05 | 32.25 | 32.39 | 32.44 | 32.48 | 32.48 | 32.48 | 32.5 | 32.5 | 32.5 |
| | EHI | **32.45** | **32.61** | **32.61** | **32.6** | **32.6** | **32.6** | **32.6** | **32.6** | **32.6** | **32.6** |
| **nDCG@5** | Exact Search | | | | | 36.03 | | | | | |
| | ScaNN | 35.38 | 35.82 | 35.91 | 35.94 | 36.01 | 36.01 | 36.03 | 36.02 | 36.03 | 36.03 |
| | Faiss-IVF | 35.56 | 35.75 | 35.89 | 35.96 | 36.0 | 36.0 | 36.02 | 36.04 | 36.04 | 36.03 |
| | EHI | **35.9** | **36.08** | **36.08** | **36.08** | **36.08** | **36.08** | **36.08** | **36.08** | **36.08** | **36.08** |
| **nDCG@10** | Exact Search | | | | | 39.39 | | | | | |
| | ScaNN | 38.61 | 39.1 | 39.25 | 39.3 | 39.37 | 39.36 | 39.38 | 39.38 | 39.39 | 39.39 |
| | Faiss-IVF | 38.86 | 39.09 | 39.23 | 39.31 | 39.35 | 39.36 | 39.36 | 39.38 | 39.38 | 39.39 |
| | EHI | **39.19** | **39.42** | **39.42** | **39.42** | **39.43** | **39.43** | **39.43** | **39.43** | **39.43** | **39.43** |
| **MRR@10** | Exact Search | | | | | 33.77 | | | | | |
| | ScaNN | 33.16 | 33.58 | 33.67 | 33.7 | 33.76 | 33.75 | 33.77 | 33.77 | 33.77 | 33.77 |
| | Faiss-IVF | 33.34 | 33.54 | 33.66 | 33.72 | 33.75 | 33.75 | 33.74 | 33.76 | 33.76 | 33.77 |
| | EHI | **33.64** | **33.81** | **33.81** | **33.8** | **33.81** | **33.8** | **33.8** | **33.8** | **33.8** | **33.8** |

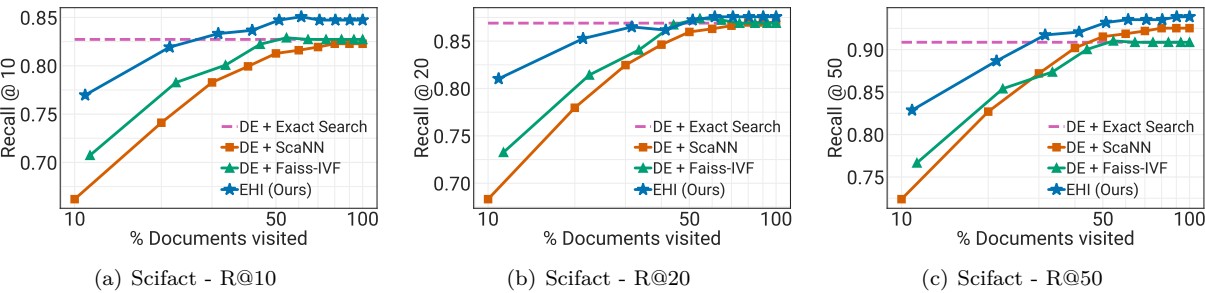

(a) Scifact - R@10       (b) Scifact - R@20       (c) Scifact - R@50

Figure 11: Additional Results for other Recall@k metrics of EHI trained on the Scifact dataset.

Table 12: Performance metrics (%) evaluated on the various datasets showcasing the effect of negative mining when finetuned from a distilbert-cos model trained on MS Marco benchmark.

| Dataset | Metric | EHI's indexer negative mining | ANCE |
|---|---|---|---|
| Scifact | R@100 | 95.27 | 95.27 |
| MS Marco (distilbert-cos) | MRR@10 | 33.8 | 33.79 |
| MS Marco (distilbert-cos) | nDCG@10 | 39.43 | 39.4 |

Furthermore, we note that most prior works, including ScaNN, DSI, and NCI, all report their numbers for a single seed. To show the robustness of our approach to initializations, we also report our numbers on multiple seeds and showcase that we significantly outperform our baselines across all the boards as further described in Section 4.4.

## C.2   Negative Mining

In this section, we discuss additional details on the exact working of our possible negative mining approach. The idea is that given positive document $(d_+)$, you would like to sample other documents which reached the same bucket as $d_+$ but are not considered relevant to the query $(q)$ per the ground truth data. Note that the dynamic negative mining is something EHI achieves for almost free, as the built indexer is also used for the ANNS. Prior approaches including Monath et al. (2023); Dahiya et al. (2023); Hofstätter et al. (2021) do study this dynamic negative sampling approaches as well. However, the difference lies in the fact that EHI does not need to build an new index for clustering every few steps, and the hierarchical tree index learned by EHI is used for the downstream ANNS task. This contrasts with works such as Monath et al. (2023); Dahiya et al. (2023); Hofstätter et al. (2021), where the index is used during training and hard negative mining, but is finally discarded. We leave this as an hyperparameter left to the practitioner. Note that we fine-tune a distilbert model which was trained on the MS Marco dataset with mined hard negatives. Thus, addition of this negative mining feature in our experiments does not lead to any significant drops as further shown in Table 12, and leads to efficient re-using of the same indexer learnt during inference.

## C.3   Comparisons against ELIAS

In this section, we compare against other end-to-end training paradigms such as ELIAS (Gupta et al., 2022) which requires warm starting of a knn index. It should be noted that ELIAS and BLISS are tailored for the domain of extreme classification while we focus on the dense information literature domain (note that label classifiers, and pre-computing KNN structure for datasets the size of MSMARCO is indeed a significant overhead). Regardless, we report our findings on the LF-AmazonTitles-131K (XC dataset) below (see Table 13). Note that below comparison is still unfair, as ELIAS utilizes additional million auxiliary parameters in their label classifiers, while EHI does not. Furthermore, EHI uses label features for encoding, while ELIAS does not. Although EHI outperforms ELIAS by 4.16% in P@1, it must be recognized that comparing against extreme classification based methods is unfair.

Table 13: Performance metrics (%) evaluated on the LF-AmazonTitles-131K dataset showcasing the efficacy of EHI in comparison to ELIAS.

| LF-AmazonTitles-131K | P@1 | P@3 | P@5 |
|---|---|---|---|
| ELIAS | 40.13 | 27.11 | 19.54 |
| EHI | 41.8 | 28.64 | 20.79 |

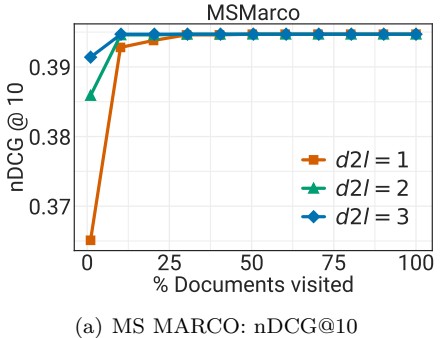
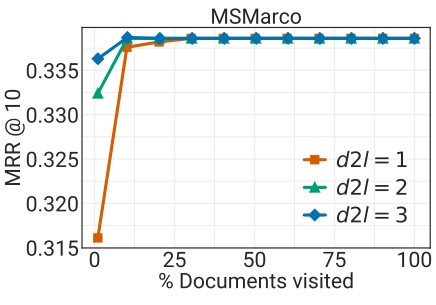

(a) MS MARCO: nDCG@10  (b) MS MARCO: MRR@10

Figure 12: Ablation study of docs to leaves on the MS Marco benchmark. Overall, we notice that as long as computational power is available, increasing number of leaves indexed for documents aids performance in retrieval.

## C.4 Allowing documents to have multiple semantics

Note that throughout training, our model works with a beam size for both query and document and tries to send relevant documents and queries to the same leaves. While testing, we have only presented results where beam size is used to send queries to multiple leaves, limiting the number of leaves per document to 1. We could use a number of leaves per document as a hyperparameter and allow documents to reach multiple leaves as long as our computational power allows.

**Accuracy:** In this section, we extend our results on increasing this factor on the MS Marco dataset as presented in Figure 12. Intuitively, each document could have multiple views, and simply sending them to only one leaf might not be ideal, but efficient indexing. Furthermore, increasing this hyperparameter from the default $d2l = 1$ setting shows strictly superior performance and improves our metrics on the MRR@10 as well as nDCG@10 metric.

**Advantages of Multiple Leaves per Document:** While assigning a document to multiple leaf nodes incurs additional memory costs, we uncover intriguing semantic properties from this approach. For the analysis, we assign multiple leaves for a document (*Doc*) ranked based on decreasing order of similarity, as shown in Figure 13. For Figure 13, we showcase three distinct examples of diverse domains showcasing not only the semantic correlation of the documents which reach similar leaves but also the queries which reach the same leaf as our example. It can be seen that each leaf captures slightly different semantics for the document. On further observations, queries mapped to the same leaf also share the semantic similarity with the *Doc*. Note that semantic similarity decreases as the document's relevance to a label decreases (left to right). This shows that EHI's training objection (Part 2 and Part 3) ensures queries with similar semantic meaning as documents and similar documents are clustered together.

**Diverse Concepts Across Leaf Nodes:** Our analysis, as showcased in Figure 14, reveals a wide range of diverse concepts captured by the word clouds derived from the leaf nodes ofEHI. For Figure 14, we visualize only a subset of randomly chosen leaves (54) instead of all leaf nodes clarity, which was obtained through random sampling. Below, we highlight some intriguing findings from our analysis. These concepts span various domains such as phones, plants, healthcare, judiciary, and more. Moreover, within each leaf node, we observe a cohesive theme among the words. For instance, the word cloud at coordinates $(1, 1)$ reflects an

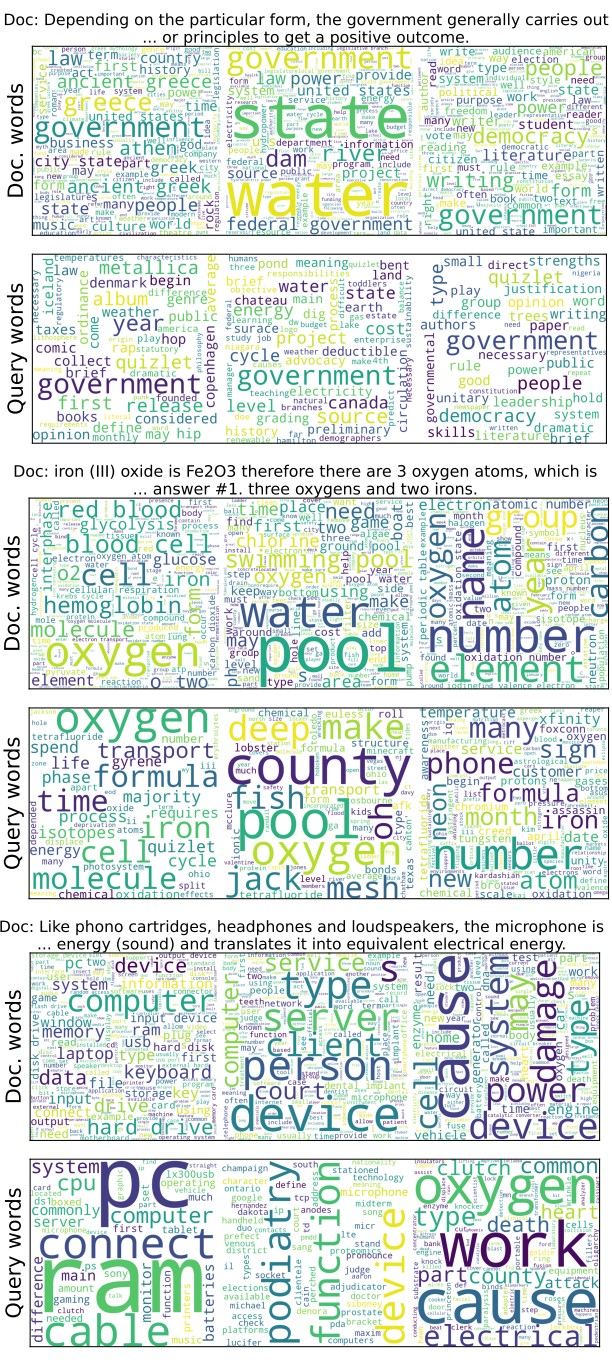

Figure 13: We deploy the concept of word cloud to summarize the semantics of all queries and documents in a leaf. Here, every document and query can hold multiple semantics therefore for the analysis, each document (*Doc*) was assigned to multiple leafs with decreasing order of similarity. It can be seen that for each leaf captures slightly different semantic for the document. On further observations, queries that were mapped to the same leaf also shares the semantic similarity with the *Doc*. Note that semantic similarity decreases as the relevance of document to a label decreases (left to right). This shows that EHI's training objection (Part 2 and Part 3) ensure queries with similar semantic meaning as document as as well as similar documents are clustered together.

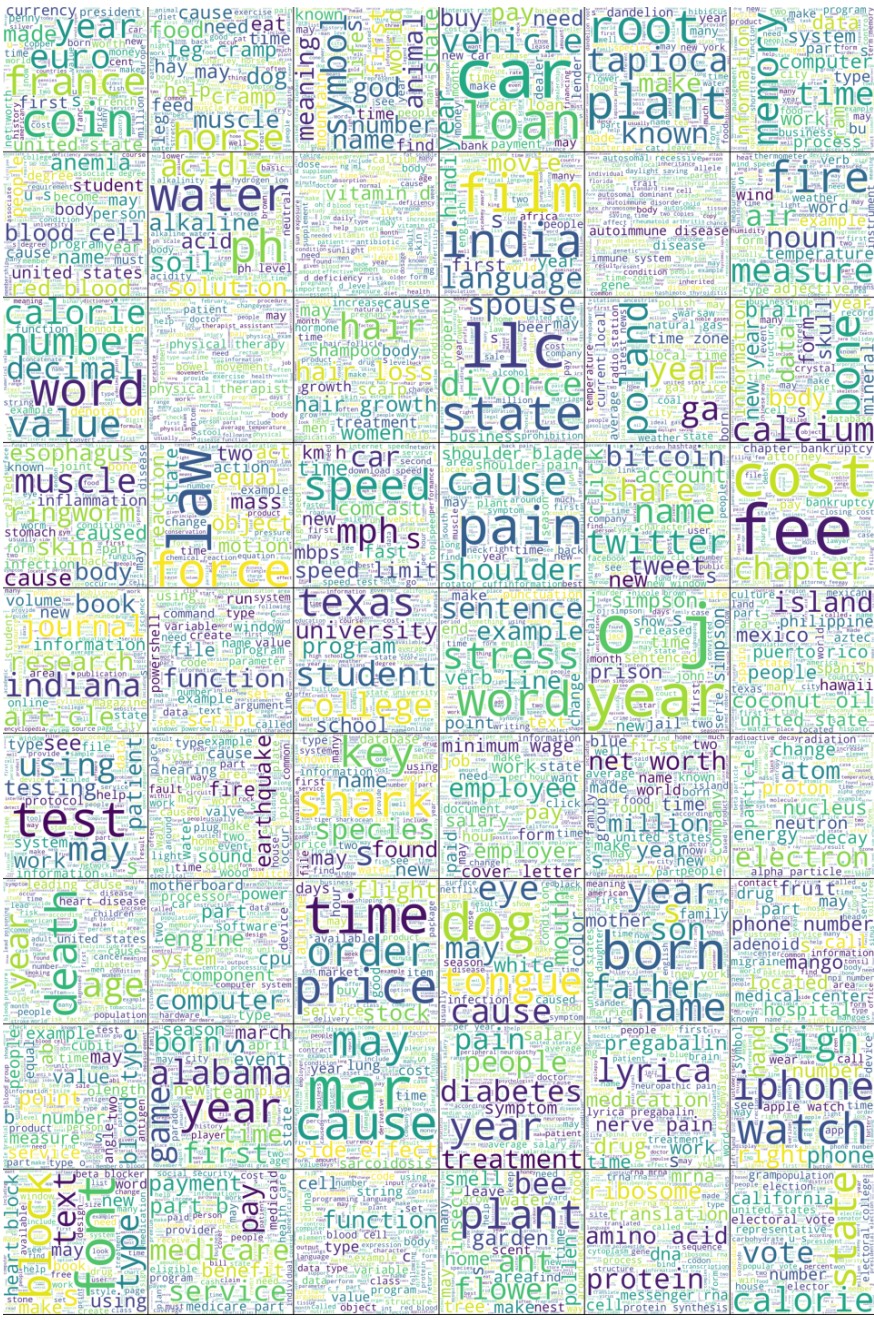

Figure 14: Figure shows that EHI's learning objective encourages leaf nodes to have different semantics. Here semantic information in each leaf is shown in the form of word cloud. The seemingly similar leaves such as (1,1), (5, 5) or (8,2) shares the similar words like "year" however, they capture completely different meaning such as financial, criminal and sports respectively.

international finance theme, while the word cloud at $(1, 2)$ pertains to fitness. This observation highlights the importance of grouping similar documents into the same leaf nodes and excluding irrelevant documents using our objective function. Note that seemingly similar leaves such as $(1, 1)$, $(5, 5)$, or $(8, 2)$ share similar words like "year"; however, they capture completely different meanings such as financial, criminal, and sports, respectively.

## D    Extending to billion-scale corpus

While this work focused on datasets up to 10M in size (MS MARCO) showing no training bottlenecks ( see Section 4), EHI holds strong theoretical advantages for scaling:

**Sublinear Retrieval Complexity** The hierarchical tree structure results in sublinear time complexity for retrieval, outperforming flat indexes as the dataset grows. This is supported by our findings in Section 4.9, where deeper trees show benefits for larger datasets.

**Efficient Search Space Partitioning** The tree efficiently partitions the search space, allowing retrieval to focus on smaller candidate subsets and significantly reducing distance computations.

Regarding the retrieval process itself, determining relevant leaf nodes for a query requires just a single forward pass through the indexer which is a standard IVF style data-structure. Final document retrieval from those leaves can leverage efficient, parallelizable matrix multiplications, similar to techniques used in billion-scale systems like ScaNN (Guo et al., 2020).

The primary computational cost lies in the initial indexing task, demanding a forward pass through the corpus. However, once indexed, retrieval becomes analogous to ScaNN, reducible to a parallelizable matrix multiplication with a (D, 768) matrix (D being the candidate document set, controlled by the beam size) and the (1, 768) query vector.

For inference, the learned index structure is equivalent to ScaNN-like data structures so the same scaling will hold (Note that even the ScaNN paper showcased its efficacy on the Glove-1.2M benchmark, while parallelizing to billions of documents in production). Furthermore, ScaNN is deployable at a billion-scale so EHI would also hold.

Training EHI is analogous to standard contrastive-style learning and poses no theoretical bottleneck in terms of document size. While our primary focus in this paper is to showcase the potential of our novel paradigm shift for dense retrieval and demonstrate its efficacy on well-tested gold benchmarks such as MS MARCO, running EHI on a billion-scale benchmark would be an interesting direction for future work. Large-scale retrieval systems like Differentiable Search Index (DSI; Tay et al. (2022)) and Neural Corpus Indexer (NCI; Wang et al. (2022)) operate in similar regimes of MS MARCO/NQ-320K, and exploring EHI's scalability in such contexts could unlock further benefits.

## E    Additional information of EHI's indexer

In this section, we hope to shed more light on the indexing process of EHI at both training and inference time. We expand upon our Figure 3, and only discuss the indexer part or the tree segment of the image in Figure 15. For a given query/document, the root node uses the embedding information ($V_{98}$ in this case) alone, and we use a linear layer ($U_1$) to transform the embeddings received by the encoder to another 768-dimensional embedding. Having these affine transformations at each height is essential as we would like to focus on different aspects of the same query/document at each height to classify which children node the item needs to go to. This is followed by another linear layer ($W_1$) to predict logits for the children at the node (2 in this case). This, followed by the softmax in the tree, leads to the probability distribution deciding which node to go towards - ($p^1 = [P_1, P_2]$). This is followed by the logic at the following height, which follows roughly the same process as the root node with one significant difference. The affine transformation at this height takes both the embedding input ($V_{98}$) and the one hot version of the probabilities at the previous height to make the predictions. During training, we have 1 at the index with the highest probability and 0 otherwise (assume $P_1 > P_2$, and hence one-hot version is $[1, 0]$). However, once the conditional probability at the next

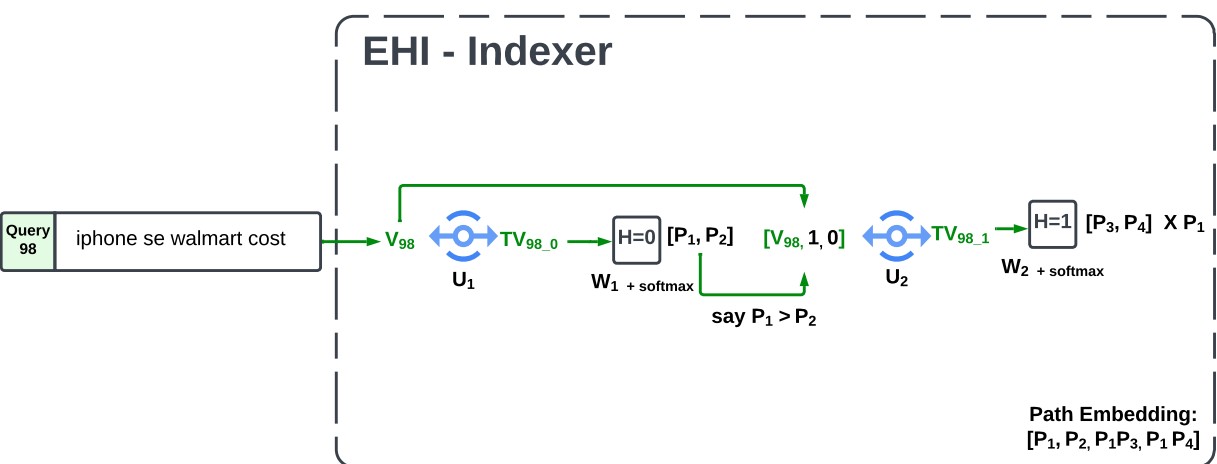

Figure 15: EHI indexer is a simple tree-structure where each height uses a neural network to. decide which path the given query or document needs to go. Following the similar paradigm as Figure 3, variables $V_{98}$ is the dense representations (embeddings) of the text of query or document.

height is computed, we multiply by this $P_1$ to have the actual probability of reaching nodes 3 and 4 (note these numberings follow the same logic as used in Figure 3). The final path embedding is the concatenation of these probability distributions leading to $[P_1, P_2, P_1P_3, P_1, P_4]$ as depicted in Figure 15. During inference, the indexer follows a similar setup where, instead of only doing one forward pass with the highest probability, we do $b$ forward passes at a given height with top-$b$ highest probabilities.

---

**Algorithm 1** Training step for EHI. $\mathbf{u}(q)$ & $\mathbf{v}(d)$ denote the query (q) & document (d) representation from encoder $\mathcal{E}_\theta$. Similarly path embedding by the indexer $(\mathcal{I}_\phi)$ is denoted by $\mathcal{T}(q)$ & $\mathcal{T}(d)$. Please refer to Algorithm 2 in appendix for the definition of `TOPK-INDEXER` and `INDEXER`. Note that the `Update(.)` function updates the encoder and indexer parameters through the back-propagation of the loss through AdamW (Loshchilov & Hutter, 2017) in our case, while other optimizers could also be used for the same (e.g. SGD, Adam, etc.).

---

**Require:** $\phi \leftarrow$ Indexer Parameters, $\theta \leftarrow$ Encoder parameters, $\tau \leftarrow$ similarity threshold, $B \leftarrow$ indexer branching factor, $H \leftarrow$ indexer height, $\beta \leftarrow$ beam size, $\mathcal{M} \leftarrow$ Mapping of leaf to documents, $\hat{\mathcal{Q}} = \{q_1, q_2, \ldots, q_s\} \sim \mathcal{Q}$ (randomly sampled mini-batch), `INDEXER` $\leftarrow$ Returns path embedding, `TOPK-INDEXER` $\leftarrow$ Returns most relevant leaf

 

    **procedure** RETRIEVER($\mathbf{u}$, $\phi$, $\beta$) – Retrieves most relevant doc
        $\_, \_, \text{Leafs} = $ `TOPK-INDEXER`$(\mathbf{u}, \phi, B, H, \beta)$                  $\triangleright$ Refer to Alg 2
        **return** $\cup_{l \in \text{Leafs}} \mathcal{M}(l)$
    **end procedure**
    **procedure** TRAINING($\hat{\mathcal{Q}}$) – jointly update encoder and indexer parameters $\theta$, $\phi$
        **loss** $= 0$
        **for** $q_i \in \hat{\mathcal{Q}}$ **do**
            $d_i \sim \{k | \mathbf{y}_{ik} = 1\}$                          $\triangleright$ Sample a relevant document
            $h_i \sim \{d | $ `RETRIEVER`$(\mathcal{E}_\theta(q_i), \phi, \beta)\} \setminus \{k | \mathbf{y}_{ik} = 1\}$     $\triangleright$ Sample a hard neg. from retriever
            **loss** $+= \mathcal{L}(\mathcal{E}_\theta(q_i), \mathcal{E}_\theta(d_i), \mathcal{E}_\theta(h_i))$          $\triangleright$ Triplet loss against hard negative
            **for** $q_j \in \hat{\mathcal{Q}}$ and $q_i \neq q_j$ **do**
                $d_j \sim \{k | \mathbf{y}_{ik} = -1 \cap \mathbf{y}_{jk} = 1\}$            $\triangleright$ Sample an in-batch negative
                **loss** $+= \mathcal{L}(\mathcal{E}_\theta(q_i), \mathcal{E}_\theta(d_i), \mathcal{E}_\theta(d_j))$       $\triangleright$ Uses $\theta$ to encode doc. and query
                **loss** $+= \mathcal{L}(\mathcal{T}_\phi(q_i), \mathcal{T}_\phi(d_i), \mathcal{T}_\phi(d_j))$         $\triangleright$ Uses `INDEXER` in Alg 2 for $\mathcal{T}_\phi$.
                **loss** $+= \mathbf{1}\{$ `SIM`$(\mathbf{v}(d_i), \mathbf{v}(d_j)) < \tau\} \mathcal{L}(\mathcal{T}_\phi(d_i), \mathcal{T}_\phi(d_i), \mathcal{T}_\phi(d_j))$
            **end for**
        **end for**
        **return** `Update`$(\theta, \phi, \textbf{loss})$         $\triangleright$ Update encoder and indexer parameters with gradient of loss
    **end procedure**

---

---

**Algorithm 2** Training step for EHI. $\mathbf{u}(q)$ & $\mathbf{v}(d)$ denote the query (q) & document (d) representation from encoder $\mathcal{E}_\theta$. Similarly path embedding by the indexer ($\mathcal{I}_\phi$) is denoted by $\mathcal{T}(q)$ & $\mathcal{T}(d)$.

---

**Require:** $\phi = \{\mathbf{W}_H, \ldots, \mathbf{W}_1, \mathbf{U}_H, \ldots, \mathbf{U}_1\} \leftarrow$ Indexer Parameters, $\theta \leftarrow$ Encoder parameters, $\tau \leftarrow$ Inter-document similarity threshold, $B \leftarrow$ branching factor of the Indexer, $H \leftarrow$ height of the indexer, $\beta \leftarrow$ beam size, $\mathcal{M} \leftarrow$ Mapping of leaf to documents, $\hat{\mathcal{Q}} = \{q_1, q_2, \ldots, q_s\} \sim \mathcal{Q}$ (randomly sampled mini-batch)

    **procedure** INDEXER($\mathbf{u}, \phi, B, H$) – Computes indexer path emb. for a query/document

        $l = \texttt{TOPK-INDEXER}(\mathbf{u}, \phi, B, H, \beta = 1)$

        $\mathcal{T}_\phi = []$

        $p_l = 1$

        **for** $h \in 1 \ldots H$ **do**

            $\hat{\mathbf{p}} = \mathcal{S}([\mathbf{o}(i_l^{h-1}); \mathbf{o}(i_l^{h-2}); \ldots; \mathbf{o}(i_l^1); \mathbf{u}]; \mathbf{W}_h, \mathbf{U}_h)$

            $\mathcal{T}_\phi = [\mathcal{T}_\phi; \hat{\mathbf{p}} \cdot p_l]$

            $p_l = p_l \cdot \hat{\mathbf{p}}[i_l^h]$

        **end for**

        **return** $\mathcal{T}_\phi$

    **end procedure**

    **procedure** TOPK-INDEXER($\mathbf{u}, \phi, B, H, \beta$) – Indexes a query/document in the $\beta$ most relevant leaves

        $h \leftarrow 1, \mathcal{P} = \{< 1, \mathbf{u}, 0 >\}$                         $\triangleright$ Tuple of score, path and node ID

        **while** $h \leq H$ **do**

            $\hat{\mathcal{P}} = \texttt{Max-Heap}(size = \beta)$

            **for** $s, \mathbf{p}, n \in \mathcal{P}.pop()$ **do**

                $\hat{\mathbf{p}} = \mathcal{S}(\mathbf{p}; \mathbf{W}_h, \mathbf{U}_h) \times s;$    For all $i \in 1 \ldots B$: $\hat{\mathcal{P}}.push(< \hat{\mathbf{p}}[i], \mathbf{o}(i); \mathbf{p}, n \cdot B + i >)$

            **end for**

            $\mathcal{P} = \hat{\mathcal{P}}$

        **end while**

        **return** $\mathcal{P}$

    **end procedure**

---

