# OpenReview forum: "EHI: End-to-end Learning of Hierarchical Index for Efficient Dense Retrieval"
_TMLR — Accepted by TMLR_

### Review · Reviewer_Nywg · 2024-07-02

**Summary Of Contributions:**

This work proposes an end-to-end hierarchical indexing (EHI) method, which overcomes the limitations of previous approximate nearest neighbor search (ANNS) methods that require separate two-stage processes for semantic search tasks. The single-stage approach of the EHI method has the advantage of reducing potential mismatches between the embedding space and the requirements of the ANNS structure. It also allows for consideration of specific characteristics of the query distribution. EHI employs a tree-structured inverted index without any initialization of cluster points, utilizing significantly compressed path embedding and a parameterized indexer. EHI is a model-agnostic method that can be integrated into various models with different encoder architectures or training schemes. Experiments on various benchmarks show that EHI achieves state-of-the-art results.

**Audience:**

Yes

**Broader Impact Concerns:**

I suppose nothing special for this paper.

**Claims And Evidence:**

Yes

**Requested Changes:**

Need explanation of what DE+ANNS stands for in section 1.2.
In Figure 1, it would be better to change e2e to end-to-end for clarification.

Overall, paper is well written and straightforward.

**Strengths And Weaknesses:**

Pros)
This paper presents the first end-to-end approach for training discrete indices-based ANNS models for text query-document retrieval. The novel tree-based architecture successfully overcomes the issues of previous two-stage approaches efficiently. Sufficient experiments are conducted to prove the advantages of the proposals. Detailed discussions on ablation results and method components are provided.

Cons)
In general, end-to-end approaches for ANNS already exist in visual domains, as mentioned in references [1-2]. It would be beneficial to clarify that this work represents the first approach specifically for the text domain.

[1] Yu, Tan, et al. "Product quantization network for fast image retrieval." Proceedings of the European Conference on Computer Vision (ECCV). 2018.
[2] Jang, Young Kyun, and Nam Ik Cho. "Generalized product quantization network for semi-supervised image retrieval." Proceedings of the IEEE/CVF Conference on Computer Vision and Pattern Recognition. 2020.

---

> ### Author Response · Authors · 2024-07-07
> **Response to Reviewer Nywg**
>
> We thank the reviewer for their positive assessment of our work and their constructive suggestions. We address their specific comments below:
>
> ---
>
> > In general, end-to-end approaches for ANNS already exist in visual domains, as mentioned in references [1-2]. It would be beneficial to clarify that this work represents the first approach specifically for the text domain.
> [1] Yu, Tan, et al. "Product quantization network for fast image retrieval." Proceedings of the European Conference on Computer Vision (ECCV). 2018. [2] Jang, Young Kyun, and Nam Ik Cho. "Generalized product quantization network for semi-supervised image retrieval." Proceedings of the IEEE/CVF Conference on Computer Vision and Pattern Recognition. 2020.
>
> - We appreciate the reviewer pointing out this important distinction. We acknowledge that end-to-end ANNS techniques have been explored in the visual domain, such as PQN [1] and GPQ [2], which primarily rely on hashing-based approaches. However, our work, EHI, introduces the first end-to-end hierarchical indexing approach specifically designed for dense text retrieval.
> EHI differentiates itself through its novel tree structure and dense path embeddings, tailored for the unique characteristics of text data and semantic search. Furthermore, EHI handles a significantly larger number of buckets (e.g., 7000) compared to the 48-bit hashing used in approaches like GPQ, demonstrating its suitability for the scale of textual datasets with millions of documents. Similarly, the scale of the experiments are also significantly larger as PQN, GPQ showcase their efficacy on ~170K images to be indexed (where the overhead of exact search might not be very large), while we experiment on ~10M documents.
>
> We have revised the manuscript to explicitly clarify this distinction between EHI and existing visual domain approaches as suggested.
>
> ---
>
> > Need explanation of what DE+ANNS stands for in section 1.2. In Figure 1, it would be better to change e2e to end-to-end for clarification.
>
> - We thank the reviewer for highlighting this point. DE+ANNS refers to the conventional two-stage approach where a standard Dual Encoder (DE) is trained independently, followed by an Approximate Nearest Neighbor Search (ANNS) using the learned embeddings. We have now clarified this explicitly in Section 1.2 for better readability. Additionally, we have expanded "e2e" to "end-to-end" in Figure 1 for improved clarity.
>
> ---
>
> We believe these changes effectively address the reviewer's valuable feedback and further strengthen the clarity and completeness of our manuscript.

---

### Review · Reviewer_y44s · 2024-07-02

**Summary Of Contributions:**

This paper introduces a novel method for dense retrieval that integrates embedding generation and approximate nearest neighbor search (ANNS) into a single, end-to-end learning framework. The proposed End-to-end Hierarchical Indexing (EHI) addresses the misalignment issues found in conventional two-stage approaches by jointly optimizing the embedding encoder and the search index structure. The authors highlight the method's capability to learn balanced and uniform clusters without needing proxy information or additional clustering algorithms. Extensive empirical evaluations demonstrate EHI's superiority over state-of-the-art techniques, showing significant improvements in retrieval accuracy on benchmarks.

**Audience:**

Yes

**Broader Impact Concerns:**

There are no broader impact concerns of this work.

**Claims And Evidence:**

Yes

**Requested Changes:**

It would be beneficial if the paper included content that addresses my opinions on its weaknesses. If such content is already included, please indicate where it can be found.

**Strengths And Weaknesses:**

## Strengths
This paper is well-written, making it easy to understand the proposed method. The motivation for the proposed approach is clear, and the method of jointly learning the encoder and ANNS is highly intriguing. The novelty of the proposed method is emphasized through a wide range of experimental results. The paper holds significant value by offering a new paradigm in information retrieval.

## Weaknesses
- The introduction of a new paradigm is highly commendable. However, several questions remain unanswered upon reading this paper:
    - Is this method applicable to "real" scenarios at a billion-scale? (While the cost of extracting dense features may be comparable to other methods, I am curious about the costs incurred during the retrieval process.)
    - If applicable, what are the computational costs?
    - When adding multiple new search documents, what procedures must be followed with the trained indexer?
- A simple question remains: What are the costs involved in hard-negative mining?
- I am curious about the reason for using distilBERT, pretrained on MS MARCO. According to the main paper and the appendix, the main contribution is the introduction of a new paradigm, but the authors would also agree that performance is crucial. Therefore, for fair comparison, it should be clear if other comparative models also used distilBERT. According to Table 4 in the appendix, only the parameters of the encoder are presented, without a detailed description of the pretrained model used.

---

> ### Author Response · Authors · 2024-07-07
> **Response to Reviewer y44s (part 1/2)**
>
> We thank the reviewer for their positive feedback and insightful questions. We address each of the raised concerns below:
>
> ---
>
> > Is this method applicable to "real" scenarios at a billion-scale? (While the cost of extracting dense features may be comparable to other methods, I am curious about the costs incurred during the retrieval process.) If applicable, what are the computational costs?
>
> - We appreciate the focus on billion-scale applicability. While this work focused on datasets up to 10M in size (MS MARCO) showing no training bottlenecks (Section 4), EHI holds strong theoretical advantages for scaling:
>
> 1. **Sublinear Retrieval Complexity:** The hierarchical tree structure results in sublinear time complexity for retrieval, outperforming flat indexes as the dataset grows. This is supported by our findings in Section 4.4, where deeper trees show benefits for larger datasets.
>
> 2. **Efficient Search Space Partitioning:** The tree efficiently partitions the search space, allowing retrieval to focus on smaller candidate subsets and significantly reducing distance computations.
>
> Regarding the retrieval process itself, determining relevant leaf nodes for a query requires just a single forward pass through the indexer which is a standard IVF style data-structure.  Final document retrieval from those leaves can leverage efficient, parallelizable matrix multiplications, similar to techniques used in billion-scale systems like ScaNN [1].
>
> The primary computational cost lies in the initial indexing task, demanding a forward pass through the corpus. However, once indexed, retrieval becomes analogous to ScaNN, reducible to a parallelizable matrix multiplication with a (D, 768) matrix (D being the candidate document set, controlled by the beam size) and the (1, 768) query vector.
>
> For inference, the learned index structure is equivalent to ScaNN-like data structures so the same scaling will hold (Note that even the ScaNN paper showcased its efficacy on the Glove-1.2M benchmark, while parallelizing to billions of documents in production).
> Furthermore, ScaNN is deployable at a billion-scale so EHI would also hold.
>
> Training EHI is analogous to standard contrastive-style learning and poses no theoretical bottleneck in terms of document size. While our primary focus in this paper is to showcase the potential of our novel paradigm shift for dense retrieval and demonstrate its efficacy on well-tested gold benchmarks such as MS MARCO, running EHI on a billion-scale benchmark would be an interesting direction for future work.  Large-scale retrieval systems like Differentiable Search Index (DSI) [2] and Neural Corpus Indexer (NCI) [3] operate in similar regimes of MS MARCO/NQ-320K, and exploring EHI's scalability in such contexts could unlock further benefits.
>
> [1] Guo, Ruiqi, et al. "Accelerating large-scale inference with anisotropic vector quantization." International Conference on Machine Learning. PMLR, 2020.
>
> [2] Tay, Yi, et al. "Transformer memory as a differentiable search index." Advances in Neural Information Processing Systems 35 (2022): 21831-21843.
>
> [3] Wang, Yujing, et al. "A neural corpus indexer for document retrieval." Advances in Neural Information Processing Systems 35 (2022): 25600-25614.
>
> ---
>
> > When adding multiple new search documents, what procedures must be followed with the trained indexer?
>
> - Indexing new documents after training is straightforward. A simple forward pass through the trained EHI network computes their path indices. These documents are then stored in the corresponding buckets. During retrieval, the process remains identical to that described in the paper, with new documents retrieved alongside the original corpus from the identified buckets. We have updated Section 3.4 to highlight the same.
>
> ---
>
> > A simple question remains: What are the costs involved in hard-negative mining?
>
> - The cost of negative mining is $\mathcal{O}(D + B)$.  *D* represents the cost of indexing each document into its corresponding bucket via EHI (a one-time cost). *B* represents the batch size used for actively sampling documents within the same bucket as the query but deemed irrelevant. Empirically, EHI training with negative mining shows a runtime increase of up to 5% compared to the exact search baseline on datasets like SciFact and FiQA.
>
> ---

---

> > ### Author Response · Authors · 2024-07-07
> > **Response to Reviewer y44s (part 2/2)**
> >
> > > using distilBERT, pretrained on MS MARCO
> >
> > - The core contribution of our work is indeed the introduction of the novel end-to-end hierarchical indexing paradigm, as showcased through the results presented in Figures 3, 5, and 6, as well as Table 1.  These experiments use a consistent setup with DistilBERT models fine-tuned from the same checkpoint to ensure an "apples-to-apples" comparison and clearly demonstrate the efficiency gains of our approach.
> >
> > - The comparison against SOTA methods in Table 4 aims to provide a broader context and demonstrate the competitiveness of EHI within the dense retrieval landscape. However, as you rightly pointed out, achieving perfectly fair comparisons in this table is challenging due to the inherent differences between existing approaches, such as encoder size, distillation techniques, late interaction mechanisms, and the use of sparse neural representations. Note that prior works including the SGPT [1], SPLADE [2] also work from a pre-trained model as training these large models are highly compute intensive.
> > To address your concern about transparency, we will add a detailed description of the pre-trained models used for each SOTA method in Table 4. We chose DistilBERT as a starting point for EHI due to its efficiency and effectiveness, allowing for faster experimentation and a clearer demonstration of the benefits of our paradigm. We acknowledge that exploring EHI with larger encoder architectures might further enhance performance, and this is an exciting direction for future work.
> >
> > - Our primary focus in this paper is to showcase the potential of our novel paradigm shift for dense retrieval, and we believe that using a consistent model architecture throughout the main experiments best serves this purpose.
> >
> > [1] Muennighoff, Niklas. "Sgpt: Gpt sentence embeddings for semantic search." arXiv preprint arXiv:2202.08904 (2022).
> >
> > [2] Formal, Thibault, et al. "From distillation to hard negative sampling: Making sparse neural ir models more effective." Proceedings of the 45th international ACM SIGIR conference on research and development in information retrieval. 2022.
> >
> > ---
> >
> > We believe that our revisions effectively address the reviewer's concerns and strengthen the clarity and completeness of our manuscript. We are confident that these changes will enhance the reader's understanding and appreciation of our work.

---

### Review · Reviewer_Qqpy · 2024-07-02

**Summary Of Contributions:**

The paper presents a hierarchical indexing algorithm for document retrieval that trains the embedding and indexing components simultaneously. Empirical evaluations show the merit of individual components of the architecture. Moreover, the the proposed method is shown the outperform several baselines on some standard benchmarks.

**Audience:**

Yes

**Claims And Evidence:**

No

**Requested Changes:**

The description and motivation of the algorithm should be considerably improved.

There should be a detailed description of the experimental set-up, and clear indication of the significance of the results. The current presentation of the experiment might be sufficient for a conference paper that highlights the promise of an algorithm, but it is insufficient for TMLR that has a strong emphasis on the trustworthiness of the presented claims.

**Strengths And Weaknesses:**

Several embedding ideas seem useful and interesting, and the combined method does appear to have strong empirical performance.

The main weakness of the paper is poor presentation. Many components are not formally defined. The description of the encoder (BERT) is dismissed with a simple citation without any further details on architecture and training. The motivation for design are very superficial, e.g., the path is computationally intractable (why would a simple path be intractable?). The training and updating is treated very superficially (e.g. by having a call to Update() in Algorithm 2).

While the experiments do appear to show improvement, there is no indication of statistical significance.

---

> ### Author Response · Authors · 2024-07-07
> **Response to Reviewer Qqpy**
>
> We thank the reviewer for their insightful comments and suggestions. We address each of their concerns below:
>
> ---
>
> > The description of the encoder (BERT) is dismissed with a simple citation without any further details on architecture and training. There should be a detailed description of the experimental set-up.
>
> - We would like to highlight that Section 3.2 does provide details on the encoder architecture. This section specifies the layers, heads, hidden dimension, and total parameters of the DistilBERT model employed in EHI. Additionally, Section 3.5 ("Training EHI") outlines the training process, including a detailed explanation of the loss function and its motivation. For completeness, Appendix C further details the hyperparameter settings, and Section 4.1 describes the broader experimental setup, including the datasets used.
>
> We believe that these details are sufficient for reproducing our results. However, we welcome any further suggestions on specific details that the reviewer finds missing.
>
> ---
>
> >  The motivation for design are very superficial, e.g., the path is computationally intractable (why would a simple path be intractable?).
>
> - We appreciate the feedback regarding the motivation behind our design choices.
> Section 3.3 explains that a naive representation of a path in our hierarchical index would necessitate a vector space with  $B^H$ dimensions, where B is the branching factor and H is the tree's height. Even for moderately sized trees, this leads to an exponential number of possible paths. For instance, with $B=10$ and $H=5$, a relatively small tree, there would be 100,000 ($10^5$) possible paths.
> - Ideally, we would like to obtain a probability distribution over every possible path for each query/document and leverage this information to: a) ensure similar items are assigned similar paths, and b) maintain a balanced distribution of documents across paths. However, computing this full probability distribution over $B^H$ paths is computationally intractable.
> - To overcome this, we introduce dense path embeddings, which can be viewed as constructing a compressed representation of a query/document's trajectory through the tree, similar to a beam search approach. This compression to a much smaller, fixed-dimensional vector space (B · H dimensions – 50 in our case) enables efficient learning and comparison of paths during training while avoiding the exponential complexity of the naive representation. We have revised the manuscript to further clarify this motivation.
>
> ---
>
> >  The training and updating is treated very superficially (e.g. by having a call to Update() in Algorithm 2).
>
> - We appreciate the feedback regarding the description of training and updating in EHI. We acknowledge that the Update() call in Algorithm 2 could benefit from a more explicit explanation.
> It is important to note that the Update($\theta$, $\phi$, loss) function in Algorithm 2 encapsulates the standard backpropagation procedure for updating the encoder ($\theta$) and indexer ($\phi$) parameters based on the calculated loss (l). This function doesn't represent a novel or complex training procedure. For instance, if the optimizer were SGD, this function would correspond to an update at step i:
>
> $\theta^{i+1} = \theta^{i} - lr * \frac{\partial L}{\partial \theta}$
>
> $\phi^{i+1} = \phi^{i} - lr * \frac{\partial L}{\partial \phi}$
>
> - Section 3.5 ("Training EHI") provides a detailed breakdown of the loss function and its motivation. This section clarifies how the loss encourages semantic similarity between relevant queries and documents, promotes consistent indexing of relevant pairs, and encourages the separation of dissimilar documents.
>
> We have revised the manuscript to explicitly clarify the role of the Update() function, avoiding any ambiguity for the reader.
>
> ---
>
> > While the experiments do appear to show improvement, there is no indication of statistical significance.
>
> - We agree that demonstrating the statistical significance of our results is crucial for establishing trustworthiness in conferences and journals alike. We disagree with the statement that statistical significance of our results are not provided.
> We would like to highlight that statistical significance tests, specifically p-tests with p=0.05, are already provided in Appendix E.3 (Table 11). We reference these results in the main paper in Section 4.2 (Page 9). Due to page limitations, we could not include the complete table in the main paper. However, we are open to suggestions on how to incorporate these findings more prominently within the allowed page count.
>
> ---
>
> We believe that our revisions effectively address the reviewer's concerns and strengthen the clarity and completeness of our manuscript. We are confident that these changes will enhance the reader's understanding and appreciation of our work.

---

### Decision · Action_Editor_4Fkc · 2024-08-08

**Recommendation:** Accept with minor revision

**Comment:**

This paper has mixed but mostly positive opinions. The main concerns include

- The end-to-end learning framework is not first proposed by this work (Reviewer Nywg)
  - This concern is addressed by revising Section 2, and focusing on "dense text retrieval"
- Comparison with SOTA -- because it uses distilBert (Reviewer y44s)
  - Table 4 was revised to include the backbone details to resolve the concern
- The presentation could be improved
  - The reason why the path is computationally intractable is now mentioned in Section 3.3.
  - In my opinion, the presentation quality still can be improved, but the current version looks okay. But, if the authors can update the manuscript, I slightly recommend revising the paper to make the motivation and experimental details more visible to novice readers. See my comment below.
- The lack of statistical significance for the experiments
  - The authors refuted this argument by citing Table 11
  - In my opinion, the current empirical evidence is somewhat okay, but the visibility of the results can be improved. See my comment below.

There were some additional comments, such as

- Practicalness of the proposed method to the billion-scale DB (Reviewer y44s)
  - As far as I understood, the discussion is not explicitly included in the revised paper. If my understanding is correct, I strongly recommend adding the related contents in the revision.

In my opinion, considering that BERT is now a very famous architecture whose detail can be omitted, the current manuscript has a weak problem with reproducibility (reproducibility can still be a problem because this submission does not include any implementation, but it is acceptable, IMO).

Overall, I think this paper can satisfy the TMLR evaluation criteria with a minor revision.

I request a minor revision by moving major contents in the Appendix to the main page. Please note that there is no page limitation for TMLR papers; the page limitation is only used to set the initial review period for the reviewers (2 weeks for less than 12 pages, 4 weeks for more than 12 pages).

In my opinion, the appendix should be an "appendix," namely, the TMRL criteria (claim and evidence) should be supported whenever the appendix's contents are ignored. For example, dataset explanations or model hyperparameter details can be in the Appendix, but many experiments that support the main claim should be in the main paper (unless an experiment has a duplicated message with any previous experiment in the main paper).

I strongly recommend moving Appendix B (Motivation for EHI) and many important experiments in Appendix D and E to the main paper. After moving the contents to the main paper, I think this paper will need a polishing stage (e.g., enhancing clarity of the manuscript), which would be a minor modification.

**Audience:**

The embedding-based retrieval is popular and actively studied in many ML applications. There will be a number of individuals interested in this paper.

**Claims And Evidence:**

### Claim

- This paper proposes a hierarchical indexing algorithm, named End-to-end Hierarchical Indexing (EHI), for information retrieval that jointly optimizes the embedding and ANN indexing.
- The main motivation is the mismatch between the embedding learning stage and the ANN indexing stage; because they are separated, the overall performance could be suboptimal compared to an  end-to-end training strategy
- EHI solves this problem with a unified optimization framework using a dual encoder for queries and documents and an inverted file index (IVF)-style tree structure

### Evidence

- This paper provides empirical evidence of the effectiveness of EHI, including
  - End-to-end framework vs. separated framework (Figure 3)
  - Design choices (Figure 4)
- The empirical evaluations were conducted on MS MARCO and TREC DL19, showing superior performances compared to the comparison methods.

---

> ### Author Response · Authors · 2024-08-29
> **Revisions for Camera-Ready draft**
>
> Dear Area Chair,
>
> Thank you for the thoughtful feedback on our manuscript, and for providing us with the opportunity to revise our submission. We have carefully addressed the concerns raised by the reviewers, and we are pleased to submit the revised camera-ready version of our paper. Below, we detail how we have responded to each of the key points mentioned in the decision:
>
> * **End-to-End Learning Framework (Reviewer Nywg)**
>
> We acknowledge that the concept of an end-to-end learning framework is not novel, as noted by the reviewer. However, our focus is on dense text retrieval, and we have revised Section 2 accordingly to clarify our contributions in this context.
>
> * **Comparison with SOTA and Use of DistilBERT (Reviewer y44s)**
>
> To address concerns regarding the use of DistilBERT, we have revised Table 4 to include detailed backbone information. This should resolve any ambiguity related to the comparison with SOTA models.
>
> * **Presentation Improvements**
>
> We have taken steps to improve the presentation of the paper, particularly by expanding Section 3.3 to include a clear explanation of why certain paths are computationally intractable. Additionally, we have aimed to improve the motivation and experimental details to enhance accessibility for novice readers, as recommended.
>
> * **Lack of Statistical Significance in Experiments**
>
> In response to the concerns about statistical significance, we have updated our discussion in Table 11 to better highlight the statistical rigor of our results. While we believe the current empirical evidence is sound, we have taken steps to increase the visibility of our results, as suggested.
>
> * **Practicality for Billion-scale DBs (Reviewer y44s)**
>
> We agree with the reviewer's suggestion and have added further discussion on the scalability of our method for billion-scale datasets. This additional content is now included in the revision, addressing the practical concerns.
>
> * **Appendix Content**
>
> As requested, we have moved key content from the Appendix (specifically Appendix B, D, and E) to the main body of the paper to ensure that the primary claims are fully supported without reliance on the Appendix. This has improved the visibility of important experiments and ensures that the main manuscript stands independently of the supplementary material.
> We hope these revisions adequately address the concerns raised, and we are confident that the revised manuscript meets the criteria for publication.
>
> Thank you once again for your time and consideration. We look forward to your feedback.